# Automatic Gradient Descent:
# Toward Deep Learning without Hyperparameters

## Abstract

Existing frameworks for deriving first-order optimisers tend to rely on generic smoothness models that are not specific to deep learning. As a prime example, the objective function is often assumed to be *Lipschitz smooth*. While these generic models do facilitate optimiser design, the resulting optimiser can be ill-suited to a given application without substantial tuning. In this paper, we develop a new framework for deriving optimisers that are directly tailored to the network architecture in question. Our framework leverages an architecture-specific smoothness model that depends explicitly on details such as the width and number of layers. Analytically minimising this smoothness model yields the *automatic gradient descent* optimiser, or AGD for short. AGD automatically sets its weight initialisation and step-size as a function of the network architecture. Our central empirical finding is that AGD successfully trains on various architectures and datasets without many of the standard optimisation hyperparameters: initialisation variance, learning rate, momentum or weight decay. A caveat to this result is that our theoretical framework works in the full batch setting, so that AGD still involves a batch size hyperparameter. Furthermore, the performance of AGD does not always match that of a highly tuned baseline. However, we suggest several directions for refining and extending our analytical framework that may resolve these issues in the future.

## 1 Introduction

Automatic differentiation has contributed to the rapid pace of innovation in the field of deep learning. Software packages such as PyTorch (Paszke et al., 2019) and Theano (Al-Rfou et al., 2016) have advanced a programming paradigm where the user (1) defines a neural network architecture by composing differentiable operators and (2) supplies training data. The package then automatically computes the gradient of the error on the training data via recursive application of the chain rule. At this point, the user becomes involved again by (3) selecting one of numerous optimisation algorithms and (4) manually tuning its hyperparameters: in particular, the learning rate (Goodfellow et al., 2016), and often other details such as the initialisation scale.

But manually tuning hyperparameters is irksome. An abundance of hyperparameters makes it difficult to rank the performance of different deep learning algorithms (Lucic et al., 2017; Schmidt et al., 2021) and difficult to reproduce results in the literature (Henderson et al., 2018). Hyperparameters confound our efforts to build a scientific understanding of generalisation in deep learning (Jiang et al., 2020; Farhang et al., 2022). And, when training neural networks at the largest scale, in pursuit of stronger forms of artificial intelligence, hyperparameter grid search can rack up millions of dollars in compute costs (Sharir et al., 2020).

Are hyperparameters just a fact of life? The thesis of this paper is that *no: they are not*. Deep learning involves fitting a known function to known data via minimising a known objective. If we could characterise these components both individually and in how they interact, then—in principle—there should be no leftover degrees of freedom to be tuned (Orabona & Cutkosky, 2020). We present a framework that attempts to remove the degrees of freedom corresponding to the initialisation and the learning rate.

Two existing tools are central to our framework, and it is their novel combination that presents the main theoretical contribution of this paper. First, a classic tool from convex analysis known as the *Bregman divergence* (Bregman, 1967; Dhillon & Tropp, 2008) is used to characterise how the neural network interacts

| Theory | Reference | Handles the Loss $\mathcal{L} \leftarrow \ell \leftarrow f$ | Non-Linear Network $f \leftarrow w$ |
|---|---|:---:|:---:|
| mirror descent | Nemirovsky & Yudin (1983) | ✓ | ✗ |
| Gauss-Newton method | Björck (1996) | ✓ | ✗ |
| natural gradient descent | Amari (1998) | ✓ | ✗ |
| neural tangent kernel | Jacot et al. (2018) | ✓ | ✗ |
| deep relative trust | Bernstein et al. (2020) | ✗ | ✓ |
| tensor programs | Yang & Hu (2021) | ✗ | ✓ |
| automatic gradient descent | this paper | ✓ | ✓ |

**Table 1: Comparing popular frameworks for first-order optimisation theory.** Frameworks differ in whether they can handle the interaction between the model output $f$ and the objective $\mathcal{L}$, and the complex non-linear interaction between the weights $w$ and the model output $f$. Our framework handles both aspects.

with the loss function. And second, a tool called *deep relative trust* (Bernstein et al., 2020) is used to characterise the highly non-linear interaction between the weights and the network output. With these tools in hand, we can apply the *majorise-minimise meta-algorithm* (Lange, 2016) to derive an optimiser explicitly tailored to deep network objective functions. We refer to this optimiser as *automatic gradient descent* (AGD). To summarise, the derivation of AGD follows three main steps:

Step 1: Functional expansion. We use a *Bregman divergence* to express the linearisation error of the objective function $\mathcal{L}(w)$ in terms of the functional perturbation $\Delta f$ to the network $f$.

Step 2: Architectural perturbation bounds. We use *deep relative trust* to relate the size and structure of the weight perturbation $\Delta w$ to the size of the induced functional perturbation $\Delta f$.

Step 3: Majorise-minimise. We substitute deep relative trust into the Bregman divergence to obtain an explicitly architecture-dependent majorisation. Minimising a slightly simplified (see Assumption 1) version of this majorisation with respect to $\Delta w$ yields an optimiser.

Empirically, we find that AGD trains a wide range of network architectures out-of-the-box, eliminating the need for a learning rate and initialisation hyperparameters. However, the theoretical framework is constructed in the full batch setting, and so when AGD is deployed in the mini-batch setting it still requires a batch size hyperparameter. And furthermore, the performance of AGD sometimes falls below that of a fully tuned baseline, both in terms of generalization and rate of convergence. For example, after training a ResNet for 300 epochs on Imagenet, there is a 10% gap in training error between SGD and AGD (Figure 4), and a 2% generalisation gap for a ResNet on CIFAR10. However, we regard the fact that AGD trains every tested architecture *quite well* as a compelling proof-of-concept that hyperparameter-free training is possible. In Section 4, we provide a number of ideas for improving and extending our framework.

Overall, we feel that the value of this paper lies in presenting a clear and coherent analytical framework for deriving deep learning optimisation algorithms, a long with a set of experimental results that demonstrate there is some merit to the framework. We believe that more work is needed to build on and refine this framework in order to obtain improved experimental results in the future.

## 1.1 Related work

**Optimisation theory** First-order optimisers leverage the first-order Taylor expansion of the objective function $\mathcal{L}(w)$—in particular, the gradient $\nabla_w \mathcal{L}(w)$. Theoretical treatments include mirror descent (Nemirovsky & Yudin, 1983), natural gradient descent (Amari, 1998) and the Gauss-Newton method (Björck, 1996). These methods have been explored in the context of deep learning (Pascanu & Bengio, 2014; Azizan & Hassibi, 2019; Sun et al., 2022). First-order methods are amenable to deep learning since the gradient of the objective is available via recursive application of the chain rule—a.k.a. error back-propagation (Rumelhart et al., 1986).

Second-order optimisers leverage the second-order Taylor expansion of the objective function $\mathcal{L}(\boldsymbol{w})$—in particular, the gradient $\nabla_{\boldsymbol{w}}\mathcal{L}(\boldsymbol{w})$ and Hessian $\nabla_{\boldsymbol{w}}^2\mathcal{L}(\boldsymbol{w})$. Examples include Newton's method (Nocedal & Wright, 1999) and cubic-regularised Newton's method (Nesterov & Polyak, 2006). Naïvely, second-order methods are less amenable to deep learning since the cost of the relevant Hessian computations is prohibitive at high dimension. That being said, efforts have been made to circumvent this issue (Agarwal et al., 2017).

The majorise-minimise meta-algorithm (Lange, 2016) is an algorithmic pattern that can be used to derive optimisers. To apply the meta-algorithm, one must first derive an upper bound on the objective which matches the objective up to $k$th-order in its Taylor series for some integer $k$. This *majorisation* can then be minimised as a proxy for reducing the original objective. Figure 1 illustrates the meta-algorithm for $k = 1$. In concurrent work (concurrent to the first online appearance of this paper) Streeter (2023) propose a class of universal majorisation-minimisation algorithms. In contrast to our approach, this form of universal majorisation-minimisation does not leverage the operator structure of the network through operator norms.

Finally, since the first online appearance of this work, two papers have built on our results. Cho & Shin (2023) find that a second-order analysis may lead to faster convergence than a full majorisation-style analysis. In a sense, a full majorisation may be too pessimistic, and this may account for why automatic gradient descent converges slower than some of our baselines. Inspired by this observation, Large et al. (2024) generalise the analysis in this paper but working to second order. However, in contrast to us, those authors do not focus on the automatic learning rate aspect and instead focus just on learning rate transfer across architectural scale.

**Deep learning theory** The *Lipschitz smoothness assumption*—a global constraint on the eigenvalues of the Hessian—is often used to derive and analyse neural network optimisers (Agarwal et al., 2016). But this assumption has been questioned (Zhang et al., 2020) and evidence has even been found for the reverse relationship, where the Hessian spectrum is highly sensitive to the choice of optimiser (Cohen et al., 2021).

These considerations motivate the development of theory that is more explicitly tailored to neural architecture. For instance, Bernstein et al. (2020) used an architectural perturbation bound termed *deep relative trust* to characterise the neural network optimisation landscape as a function of network depth. Similarly, Yang & Hu (2021) sought to understand the role of width, leading to their *maximal update parameterisation*.

## 1.2 Preliminaries

Given a vector $\boldsymbol{v}$ in $\mathbb{R}^n$, we will need to measure its size in three different ways:

**Definition 1** (Manhattan norm) The *Manhattan norm* $\|\cdot\|_1$ of a vector $\boldsymbol{v}$ is defined by $\|\boldsymbol{v}\|_1 := \sum_i |\boldsymbol{v}_i|$.

**Definition 2** (Euclidean norm) The *Euclidean norm* $\|\cdot\|_2$ of a vector $\boldsymbol{v}$ is defined by $\|\boldsymbol{v}\|_2 := \sqrt{\sum_i \boldsymbol{v}_i^2}$.

**Definition 3** (Infinity norm) The *infinity norm* $\|\cdot\|_\infty$ of a vector $\boldsymbol{v}$ is defined by $\|\boldsymbol{v}\|_\infty := \max_i |\boldsymbol{v}_i|$.

For a matrix $\boldsymbol{M}$ in $\mathbb{R}^{m\times n}$, the reader should be aware that it has a singular value decomposition:

**Fact 1** (SVD) Every matrix $\boldsymbol{M}$ in $\mathbb{R}^{m\times n}$ admits a *singular value decomposition* (SVD) of the form $\boldsymbol{M} = \sum_{i=1}^{\min(m,n)} \sigma_i(\boldsymbol{M}) \cdot \boldsymbol{u}_i\boldsymbol{v}_i^\top$ where the *left singular vectors* $\{\boldsymbol{u}_i\}$ are orthonormal vectors in $\mathbb{R}^m$, the *right singular vectors* $\{\boldsymbol{v}_i\}$ are orthonormal vectors in $\mathbb{R}^m$ and the *singular values* $\{\sigma_i(\boldsymbol{M})\}$ are non-negative scalars.

The singular value decomposition allows us to measure the size of a matrix in two different ways:

**Definition 4** (Frobenius norm) The *Frobenius norm* $\|\cdot\|_F$ of a matrix $\boldsymbol{M}$ is given by $\|\boldsymbol{M}\|_F := \sqrt{\sum_i \sigma_i(\boldsymbol{M})^2}$.

**Definition 5** (Operator norm) The *operator norm* $\|\cdot\|_*$ of a matrix $\boldsymbol{M}$ is given by $\|\boldsymbol{M}\|_* := \max_i \sigma_i(\boldsymbol{M})$.

While the operator norm $\|\boldsymbol{M}\|_*$ reports the largest singular value, the quantity $\|\boldsymbol{M}\|_F/\sqrt{\min(m,n)}$ reports the root mean square singular value. Finally, we will need to understand two aspects of matrix conditioning:

**Definition 6** (Rank) The *rank* of a matrix counts the number of non-zero singular values.

**Definition 7** (Stable rank) The *stable rank* of a matrix $\boldsymbol{M}$ is defined by $\operatorname{rank}_{\text{stable}} \boldsymbol{M} := \|\boldsymbol{M}\|_F^2/\|\boldsymbol{M}\|_*^2$.

The stable rank provides an approximation to the rank that ignores the presence of very small singular values. Let us consider the extremes. An orthogonal matrix $\boldsymbol{O} \in \mathbb{R}^{m\times n}$ has both full rank and full stable rank: $\operatorname{rank}\boldsymbol{O} = \operatorname{rank}_{\text{stable}}\boldsymbol{O} = \min(m,n)$. A rank-one matrix $\boldsymbol{P}$ has unit stable rank and satisfies $\|\boldsymbol{P}\|_* = \|\boldsymbol{P}\|_F$.

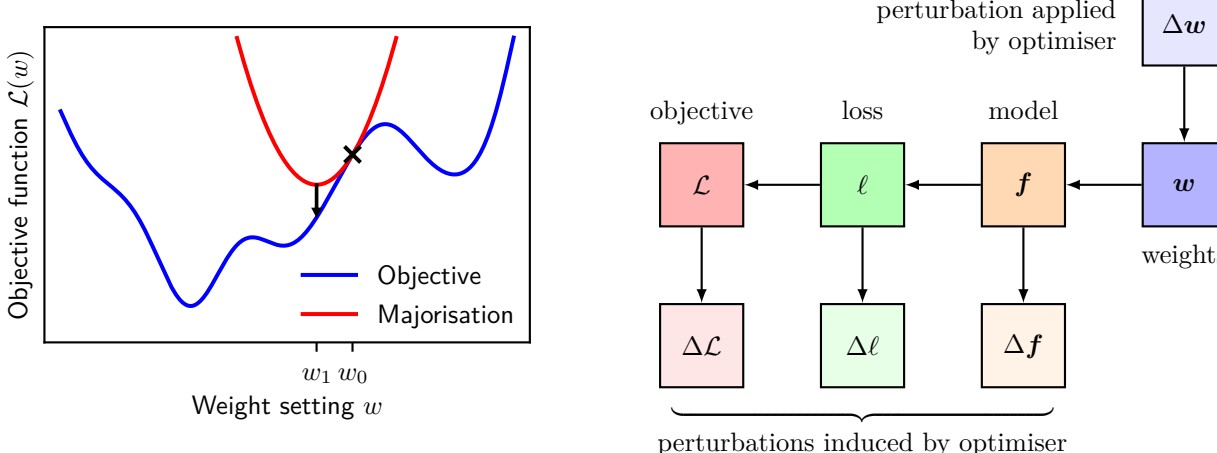

**Figure 1: Majorise-minimise and the perturbation hierarchy.** The **left panel** depicts the majorise-minimise meta-algorithm (Lange, 2016), which is an algorithmic pattern for reducing an objective (blue) by minimising a sequence of upper bounds (one shown in red). The upper bounds, known as a *majorisation*, must lie tangent to the objective to guarantee an improvement in one step of the meta-algorithm. The **right panel** depicts the perturbation hierarchy of a generic machine learning model: the optimiser perturbs the weights and this induces perturbations to the model output, the loss on individual training examples and ultimately the overall objective. Majorising machine learning objective functions requires addressing the full perturbation hierarchy.

## 2 Majorise-Minimise for Generic Learning Problems

This section develops a framework for applying the majorise-minimise meta-algorithm to generic optimisation problems in machine learning. In particular, the novel technique of *functional expansion* is introduced. Section 3 will specialise this technique to deep neural networks. All proofs are supplied in Appendix B.

Given a machine learning model and a set of training data, our objective is to minimise the error of the model, averaged over the training data. Formally, we would like to minimise the following function:

**Definition 8** (Composite objective) Consider a machine learning model $\boldsymbol{f}$ that maps an input $\boldsymbol{x}$ and a weight vector $\boldsymbol{w}$ to output $\boldsymbol{f}(\boldsymbol{x};\boldsymbol{w})$. Given data $\mathsf{S}$ and a convex loss function $\ell$, the *objective* $\mathcal{L}(\boldsymbol{w})$ is defined by:

$$\mathcal{L}(\boldsymbol{w}) := \frac{1}{|\mathsf{S}|} \sum_{(\boldsymbol{x},\boldsymbol{y})\in\mathsf{S}} \ell(\boldsymbol{f}(\boldsymbol{x};\boldsymbol{w}),\boldsymbol{y}).$$

We refer to this objective as *composite* since the loss function $\ell$ is *composed* with a machine learning model $\boldsymbol{f}$. While the loss function itself is convex, the overall composite is often non-convex due to the non-linear machine learning model. Common convex loss functions include the square loss and the cross-entropy loss:

**Example 1** (Square loss) The *square loss* is defined by: $\ell(\boldsymbol{f}(\boldsymbol{x};\boldsymbol{w}),\boldsymbol{y}) := \frac{1}{2d_L}\|\boldsymbol{f}(\boldsymbol{x};\boldsymbol{w}) - \boldsymbol{y}\|_2^2$.

**Example 2** (Xent loss) The *cross-entropy (xent) loss* is defined by: $\ell(\boldsymbol{f}(\boldsymbol{x}),\boldsymbol{y}) := -\log[\mathrm{softmax}(\boldsymbol{f}(\boldsymbol{x}))]^\top\boldsymbol{y}$, where the softmax function is defined by $\mathrm{softmax}(\boldsymbol{f}(\boldsymbol{x})) := \exp\boldsymbol{f}(\boldsymbol{x})/\|\exp\boldsymbol{f}(\boldsymbol{x})\|_1$.

### 2.1 Decomposition of linearisation error

First-order optimisers leverage the linearisation of the objective at the current iterate. To design such methods, we must understand the realm of validity of this linearisation. To that end, we derive a general decomposition of the linearisation error of a machine learning system. The result is stated in terms of a *perturbation hierarchy*. In particular, perturbing the weight vector of a machine learning model $\boldsymbol{w} \to \boldsymbol{w} + \Delta\boldsymbol{w}$ induces perturbations to the model output $\boldsymbol{f} \to \boldsymbol{f} + \Delta\boldsymbol{f}$, to the loss on individual data samples $\ell \to \ell + \Delta\ell$

and, at last, to the overall objective function $\mathcal{L} \to \mathcal{L} + \Delta\mathcal{L}$. Formally, a weight perturbation $\Delta \boldsymbol{w}$ induces:

$$\Delta\boldsymbol{f}(\boldsymbol{x}) \qquad := \boldsymbol{f}(\boldsymbol{x}; \boldsymbol{w} + \Delta\boldsymbol{w}) - \boldsymbol{f}(\boldsymbol{x}; \boldsymbol{w}); \qquad\qquad\qquad \text{(functional perturbation)}$$
$$\Delta\ell(\boldsymbol{f}(\boldsymbol{x}), \boldsymbol{y}) \quad := \ell(\boldsymbol{f}(\boldsymbol{x}) + \Delta\boldsymbol{f}(\boldsymbol{x}), \boldsymbol{y}) - \ell(\boldsymbol{f}(\boldsymbol{x}), \boldsymbol{y}); \qquad\qquad \text{(loss perturbation)}$$
$$\Delta\mathcal{L}(\boldsymbol{w}) \qquad := \frac{1}{|\mathsf{S}|} \sum_{(\boldsymbol{x},\boldsymbol{y})\in\mathsf{S}} \Delta\ell(\boldsymbol{f}(\boldsymbol{x}), \boldsymbol{y}). \qquad\qquad\qquad \text{(objective perturbation)}$$

We have adopted a compact notation where the dependence of $\boldsymbol{f}(\boldsymbol{x}; \boldsymbol{w})$ on $\boldsymbol{w}$ is at times suppressed. The perturbation hierarchies of a generic machine learning model and a deep neural network are visualised in Figures 1 and 2, respectively. The linearisation error of the objective perturbation $\Delta\mathcal{L}$ decomposes as:

**Proposition 1** (Decomposition of linearisation error) For any differentiable loss $\ell$ and any differentiable machine learning model $\boldsymbol{f}$ the linearisation error of the objective function $\mathcal{L}$ admits the following decomposition:

$$\underbrace{\Delta\mathcal{L}(\boldsymbol{w}) - \nabla_{\boldsymbol{w}}\mathcal{L}(\boldsymbol{w})^{\top}\Delta\boldsymbol{w}}_{\text{linearisation error of objective}} \quad = \quad \frac{1}{|\mathsf{S}|} \sum_{(\boldsymbol{x},\boldsymbol{y})\in\mathsf{S}} \nabla_{\boldsymbol{f}(\boldsymbol{x})}\ell(\boldsymbol{f}(\boldsymbol{x}), \boldsymbol{y})^{\top} \underbrace{[\Delta\boldsymbol{f}(\boldsymbol{x}) - \nabla_{\boldsymbol{w}}\boldsymbol{f}(\boldsymbol{x})\Delta\boldsymbol{w}]}_{\text{linearisation error of model}}$$
$$+ \frac{1}{|\mathsf{S}|} \sum_{(\boldsymbol{x},\boldsymbol{y})\in\mathsf{S}} \underbrace{\Delta\ell(\boldsymbol{f}(\boldsymbol{x}), \boldsymbol{y}) - \nabla_{\boldsymbol{f}(\boldsymbol{x})}\ell(\boldsymbol{f}(\boldsymbol{x}), \boldsymbol{y})^{\top}\Delta\boldsymbol{f}(\boldsymbol{x})}_{\text{linearisation error of loss}}.$$

In words: the linearisation error of the objective decomposes into two terms. The first term depends on the linearisation error of the machine learning model and the second term the linearisation error of the loss. This decomposition relies on nothing but differentiability. Proposition 1 may also be seen as a generalisation of the Gauss-Newton decomposition of the Hessian that holds to all orders. For a convex loss, the second term may be interpreted as a Bregman divergence:

**Definition 9** (Bregman divergence of loss) For any convex loss $\ell$:

$$\mathrm{bregman}_{\ell(\cdot,\boldsymbol{y})}(\boldsymbol{f}(\boldsymbol{x}), \Delta\boldsymbol{f}(\boldsymbol{x})) := \Delta\ell(\boldsymbol{f}(\boldsymbol{x}), \boldsymbol{y}) - \nabla_{\boldsymbol{f}(\boldsymbol{x})}\ell(\boldsymbol{f}(\boldsymbol{x}), \boldsymbol{y})^{\top}\Delta\boldsymbol{f}(\boldsymbol{x}).$$

A Bregman divergence is just the linearisation error of a convex function. Two important examples are:

**Lemma 1** (Bregman divergence of square loss) When $\ell$ is set to square loss, then:

$$\mathrm{bregman}_{\ell(\cdot,\boldsymbol{y})}(\boldsymbol{f}(\boldsymbol{x}), \Delta\boldsymbol{f}(\boldsymbol{x})) = \frac{1}{2d_L}\|\Delta\boldsymbol{f}(\boldsymbol{x})\|_2^2.$$

**Lemma 2** (Bregman divergence of xent loss) When $\ell$ is set to cross-entropy loss, and if $\boldsymbol{y}^{\top}\boldsymbol{1} = 1$, then:

$$\mathrm{bregman}_{\ell(\cdot,\boldsymbol{y})}(\boldsymbol{f}(\boldsymbol{x}), \Delta\boldsymbol{f}(\boldsymbol{x})) = D_{\mathrm{KL}}\Big(\mathrm{softmax}(\boldsymbol{f}(\boldsymbol{x})) \,\Big\|\, \mathrm{softmax}(\boldsymbol{f}(\boldsymbol{x}) + \Delta\boldsymbol{f}(\boldsymbol{x}))\Big)$$
$$\leqslant \frac{1}{2}\|\Delta\boldsymbol{f}(\boldsymbol{x})\|_\infty^2 + \mathcal{O}(\Delta\boldsymbol{f}^3).$$

Our methods may be applied to other convex losses by calculating or bounding their Bregman divergence.

## 2.2 Functional expansion and functional majorisation

Before continuing, we make one simplifying assumption. Observe that the first term on the right-hand side of Proposition 1 is a high-dimensional inner product between two vectors. Since there is no clear reason why these two vectors should be aligned, let us assume for convenience that their inner product is zero:

**Assumption 1** (Orthogonality of model linearisation error) In the same setting as Proposition 1:

$$\frac{1}{|\mathsf{S}|} \sum_{(\boldsymbol{x},\boldsymbol{y})\in\mathsf{S}} \nabla_{\boldsymbol{f}(\boldsymbol{x})}\ell(\boldsymbol{f}(\boldsymbol{x}), \boldsymbol{y})^{\top} \underbrace{[\Delta\boldsymbol{f}(\boldsymbol{x}) - \nabla_{\boldsymbol{w}}\boldsymbol{f}(\boldsymbol{x})\Delta\boldsymbol{w}]}_{\text{linearisation error of model}} = 0.$$

While one can work without this assumption in special cases (Bernstein, 2022), we found that its inclusion simplifies the analysis and in practice did not lead to a discernible weakening of the resulting algorithm. In any case, this assumption is considerably milder than the common assumption in the literature (Pascanu & Bengio, 2014; Lee et al., 2019) that the model linearisation error is itself zero: $[\Delta\boldsymbol{f}(\boldsymbol{x}) - \nabla_{\boldsymbol{w}}\boldsymbol{f}(\boldsymbol{x})\Delta\boldsymbol{w}] = 0$.

Armed with Proposition 1 and Assumption 1, we are ready to introduce functional expansion and majorisation:

**Theorem 1** (Functional expansion) Consider a convex differentiable loss $\ell$ and a differentiable machine learning model $\boldsymbol{f}$. Under Assumption 1, the corresponding composite objective $\mathcal{L}$ admits the expansion:

$$\mathcal{L}(\boldsymbol{w} + \Delta\boldsymbol{w}) = \underbrace{\mathcal{L}(\boldsymbol{w}) + \nabla_{\boldsymbol{w}}\mathcal{L}(\boldsymbol{w})^\top \Delta\boldsymbol{w}}_{\text{first-order Taylor series}} + \frac{1}{|\mathsf{S}|} \sum_{(\boldsymbol{x},\boldsymbol{y})\in\mathsf{S}} \mathrm{bregman}_{\ell(\cdot,\boldsymbol{y})}(\boldsymbol{f}(\boldsymbol{x}), \Delta\boldsymbol{f}(\boldsymbol{x})).$$

So the perturbed objective $\mathcal{L}(\boldsymbol{w} + \Delta\boldsymbol{w})$ may be written as the sum of its first-order Taylor expansion with a Bregman divergence in the model outputs averaged over the training set. It is straightforward to specialise this result to different losses by substituting in their Bregman divergence:

**Corollary 1** (Functional expansion of mean squared error) Under Assumption 1, for square loss:

$$\mathcal{L}(\boldsymbol{w} + \Delta\boldsymbol{w}) = \mathcal{L}(\boldsymbol{w}) + \nabla_{\boldsymbol{w}}\mathcal{L}(\boldsymbol{w})^\top \Delta\boldsymbol{w} + \frac{1}{|\mathsf{S}|} \sum_{(\boldsymbol{x},\boldsymbol{y})\in\mathsf{S}} \frac{1}{2d_L}\|\Delta\boldsymbol{f}(\boldsymbol{x})\|_2^2.$$

**Corollary 2** (Functional majorisation for xent loss) Under Assumption 1, for cross-entropy loss, if $\boldsymbol{y}^\top \mathbf{1} = 1$:

$$\mathcal{L}(\boldsymbol{w} + \Delta\boldsymbol{w}) \leqslant \mathcal{L}(\boldsymbol{w}) + \nabla_{\boldsymbol{w}}\mathcal{L}(\boldsymbol{w})^\top \Delta\boldsymbol{w} + \frac{1}{|\mathsf{S}|} \sum_{(\boldsymbol{x},\boldsymbol{y})\in\mathsf{S}} \frac{1}{2}\|\Delta\boldsymbol{f}(\boldsymbol{x})\|_\infty^2 + \mathcal{O}(\Delta\boldsymbol{f}^3).$$

When the functional perturbation is reasonably "spread out", we would expect $\|\Delta\boldsymbol{f}(\boldsymbol{x})\|_\infty^2 \approx \|\Delta\boldsymbol{f}(\boldsymbol{x})\|_2^2/d_L$. In this setting, the functional majorisation of cross-entropy loss agrees with the functional expansion of mean squared error to second order. While the paper derives automatic gradient descent for the square loss, this observation justifies its application to cross-entropy loss, as in the case of the ImageNet experiments.

## 2.3 Recovering existing frameworks

We briefly observe that three existing optimisation frameworks may be recovered efficiently from Theorem 1:

**Mirror descent** For linear models $\boldsymbol{f}(\boldsymbol{x}; \boldsymbol{W}) := \boldsymbol{W}\boldsymbol{x}$, the Bregman divergence $\mathrm{bregman}_{\ell(\cdot,\boldsymbol{y})}(\boldsymbol{f}(\boldsymbol{x}), \Delta\boldsymbol{f}(\boldsymbol{x}))$ may be written $\mathrm{bregman}_{\ell(\cdot,\boldsymbol{y})}(\boldsymbol{W}\boldsymbol{x}, \Delta\boldsymbol{W}\boldsymbol{x})$. This is a convex function of the weight perturbation $\Delta\boldsymbol{W}$. Substituting into Theorem 1 and minimising with respect to $\Delta\boldsymbol{W}$ is the starting point for mirror descent.

**Gauss-Newton method** Substituting the linearised functional perturbation $\Delta\boldsymbol{f}(\boldsymbol{x}) \approx \nabla_{\boldsymbol{w}}\boldsymbol{f}(\boldsymbol{x})\Delta\boldsymbol{w}$ into Corollary 1 and minimising with respect to $\Delta\boldsymbol{w}$ is the starting point for the Gauss-Newton method.

**Natural gradient descent** Substituting the linearised functional perturbation $\Delta\boldsymbol{f}(\boldsymbol{x}) \approx \nabla_{\boldsymbol{w}}\boldsymbol{f}(\boldsymbol{x})\Delta\boldsymbol{w}$ into Corollary 2 and minimising with respect to $\Delta\boldsymbol{w}$ is the starting point for natural gradient descent.

# 3 Majorise-Minimise for Deep Learning Problems

In this section, we will focus our efforts on deriving an optimiser for deep fully-connected networks trained with square loss. The derivation for cross-entropy loss is analogous. Proofs are relegated to Appendix B.

**Definition 10** (Fully-connected network) A *fully-connected network (FCN)* $\boldsymbol{f}$ of depth $L$ maps an input $\boldsymbol{x} \in \mathbb{R}^{d_0}$ to an output $\boldsymbol{f}(\boldsymbol{x}; \boldsymbol{w}) \in \mathbb{R}^{d_L}$ via $L$ matrix multiplications interspersed by non-linearity $\mathrm{relu}(z) := \max(0, z)$:

$$\boldsymbol{f}(\boldsymbol{x}; \boldsymbol{w}) := \boldsymbol{W}_L \circ (\mathrm{relu} \circ \boldsymbol{W}_{L-1}) \circ (\mathrm{relu} \circ \boldsymbol{W}_{L-2}) \circ \cdots \circ (\mathrm{relu} \circ \boldsymbol{W}_1\boldsymbol{x}).$$

In this expression, $\boldsymbol{w}$ denotes the tuple of matrices $\boldsymbol{w} = (\boldsymbol{W}_1, ..., \boldsymbol{W}_L)$ with $k$th matrix $\boldsymbol{W}_k$ in $\mathbb{R}^{d_k \times d_{k-1}}$. In what follows, we will find the following dimensional scaling to be particularly convenient:

**Prescription 1** (Dimensional scaling) For $\eta > 0$, the data $(\boldsymbol{x}, \boldsymbol{y})$, weights $\boldsymbol{W}_k$ and updates $\Delta\boldsymbol{W}_k$ should obey:

$$\|\boldsymbol{x}\|_2 = \sqrt{d_0}; \qquad\qquad\qquad \text{(input scaling)}$$
$$\|\boldsymbol{W}_k\|_* = \sqrt{d_k/d_{k-1}} \qquad \text{for all } k = 1, ..., L; \qquad \text{(weight scaling)}$$
$$\|\Delta\boldsymbol{W}_k\|_* = \sqrt{d_k/d_{k-1}} \cdot \frac{\eta}{L} \qquad \text{for all } k = 1, ..., L; \qquad \text{(update scaling)}$$
$$\|\boldsymbol{y}\|_2 = \sqrt{d_L}. \qquad\qquad\qquad \text{(target scaling)}$$

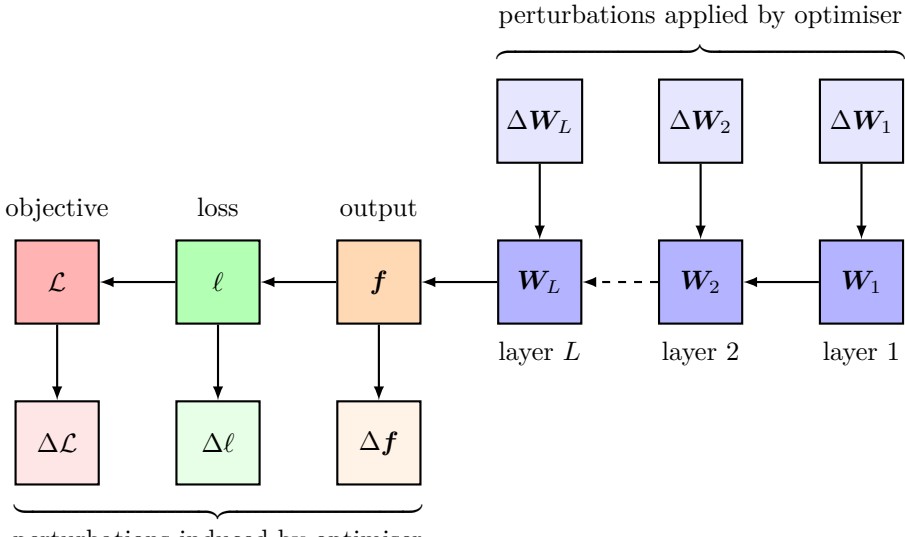

**Figure 2: Perturbation hierarchy of a deep neural network.** When training a neural network, the optimiser applies structured perturbations to the weights, in the form of one perturbation matrix $\Delta \boldsymbol{W}_k$ per weight matrix $\boldsymbol{W}_k$. Deep relative trust (Bernstein et al., 2020) provides a tool to understand how structured weight perturbations of this form affect the network output $\boldsymbol{f}$. Combining deep relative trust with a Bregman divergence (Bregman, 1967) allows us to analyse the full perturbation hierarchy.

This prescription has subsequently been shown to permit feature learning in neural networks for fixed step size $\eta$ at varying width (Yang et al., 2023) unlike the PyTorch default parameterisation. While results can be derived without adopting Prescription 1, the scalings substantially simplify our formulae. One reason for this is that, under Prescription 1, we have the telescoping property that $\prod_{k=1}^{L} \|\boldsymbol{W}_k\|_* = \sqrt{d_L/d_0}$. For a concrete example of how this helps, consider the following bound on the norm of the network outputs:

**Lemma 3** (Output bound) The output norm of a fully-connected network $\boldsymbol{f}$ obeys the following bound:

$$\|\boldsymbol{f}(\boldsymbol{x}; \boldsymbol{w})\|_2 \leqslant \left[\prod_{k=1}^{L} \|\boldsymbol{W}_k\|_*\right] \times \|\boldsymbol{x}\|_2 = \sqrt{d_L} \text{ under Prescription 1.}$$

So, under Prescription 1, the bound is simple. Furthermore, the scaling of the update with a single parameter $\eta$ reduces the problem of solving for an optimiser to a single parameter problem. To see how this might make life easier, consider the following lemma that relates weight perturbations to functional perturbations:

**Lemma 4** (Deep relative trust) When adjusting the weights $\boldsymbol{w} = (\boldsymbol{W}_1, ..., \boldsymbol{W}_L)$ of a fully-connected network $\boldsymbol{f}$ by $\Delta \boldsymbol{w} = (\Delta \boldsymbol{W}_1, ..., \Delta \boldsymbol{W}_L)$, the induced functional perturbation $\Delta \boldsymbol{f}(\boldsymbol{x}) := \boldsymbol{f}(\boldsymbol{x}; \boldsymbol{w} + \Delta \boldsymbol{w}) - \boldsymbol{f}(\boldsymbol{x}; \boldsymbol{w})$ obeys:

$$\|\Delta \boldsymbol{f}(\boldsymbol{x})\|_2 \leqslant \left[\prod_{k=1}^{L} \|\boldsymbol{W}_k\|_*\right] \times \|\boldsymbol{x}\|_2 \times \left[\prod_{k=1}^{L} \left(1 + \frac{\|\Delta \boldsymbol{W}_k\|_*}{\|\boldsymbol{W}_k\|_*}\right) - 1\right] \leqslant \sqrt{d_L} \times (\exp \eta - 1) \text{ under Prescription 1.}$$

So, under Prescription 1, the single parameter $\eta$ directly controls the size of functional perturbations.

In terms of enforcing Prescription 1 in practice, the norms of the data $(\boldsymbol{x}, \boldsymbol{y})$ may be set via pre-processing, the norm of the update $\Delta \boldsymbol{W}_k$ may be set via the optimisation algorithm and the norm of the weight matrix $\boldsymbol{W}_k$ may be set by the choice of initialisation. While, yes, $\|\boldsymbol{W}_k\|_*$ may drift during training, the amount that this can happen is limited by Weyl (1912)'s inequality for singular values. In particular, after one step the perturbed operator norm $\|\boldsymbol{W}_k + \Delta \boldsymbol{W}_K\|_*$ is sandwiched like $(1 - \eta/L) \cdot \|\boldsymbol{W}_k\|_* \leqslant \|\boldsymbol{W}_k + \Delta \boldsymbol{W}_K\|_* \leqslant (1 + \eta/L) \cdot \|\boldsymbol{W}_k\|_*$. While this drift may accumulate over steps, some form of projection could be used to correct for this, but we defer the empirical study of this kind of projection to future work.

```
def initialise_weights():

    for layer k in {1,...,L}:

        W_k ~ UNIFORM(orthogonal(d_k, d_{k-1}))        # sample a semi-orthogonal matrix

        W_k ← W_k · √(d_k/d_{k-1})                      # rescale its singular values

def update_weights():

    G ← (1/L) Σ_{l=1}^{L} ‖∇_{W_k}L‖_F · √(d_k/d_{k-1})    # get gradient summary

    η ← log (1+√(1+4G))/2                              # set automatic learning rate

    for layer k in {1,...,L}:

        W_k ← W_k - (η/L) · (∇_{w_k}L / ‖∇_{W_k}L‖_F) · √(d_k/d_{k-1})   # update weights
```

**Algorithm 1: Automatic gradient descent.** The matrix $\boldsymbol{W}_k$ in $\mathbb{R}^{d_k \times d_{k-1}}$ is the weight matrix at layer $k$. The gradient $\nabla_{\boldsymbol{W}_k}\mathcal{L}$ is with respect to the objective $\mathcal{L}$ evaluated on a mini-batch $B$ of training samples.

### 3.1 Deriving automatic gradient descent

With both functional majorisation and deep relative trust in hand, we can majorise the deep network objective:

**Lemma 5** (Exponential majorisation) For an FCN with square loss, under Assumption 1 and Prescription 1:

$$\mathcal{L}(\boldsymbol{w} + \Delta\boldsymbol{w}) \leqslant \mathcal{L}(\boldsymbol{w}) + \frac{\eta}{L} \sum_{k=1}^{L} \left[ \sqrt{d_k/d_{k-1}} \times \mathrm{tr}\frac{\Delta\boldsymbol{W}_k^\top \nabla_{\boldsymbol{W}_k}\mathcal{L}}{\|\Delta\boldsymbol{W}_k\|_*} \right] + \tfrac{1}{2}\left(\exp\eta - 1\right)^2.$$

Observe that the majorisation only depends on the magnitude of the scalar $\eta$ and on some notion of alignment $\mathrm{tr}\,\Delta\boldsymbol{W}_k^\top\nabla_{\boldsymbol{W}_k}\mathcal{L}/\|\Delta\boldsymbol{W}_k\|_*$ between the perturbation matrix $\Delta\boldsymbol{W}_k$ and the gradient matrix $\nabla_{\boldsymbol{W}_k}\mathcal{L}$. To derive an optimiser, we would now like to minimise this majorisation with respect to $\eta$ and this angle. First, let us introduce one additional assumption and one additional definition:

**Assumption 2** (Gradient conditioning) The gradient satisfies $\mathrm{rank}_{\mathrm{stable}}\nabla_{\boldsymbol{W}_k}\mathcal{L} = 1$ at all layers $k = 1,...,L$.

This assumption implies that the Frobenius norm $\|\nabla_{\boldsymbol{W}_k}\mathcal{L}\|_F$ and operator norm $\|\nabla_{\boldsymbol{W}_k}\mathcal{L}\|_*$ of the gradient at layer $k$ are equal. It is not immediately obvious why this should be a good assumption. After all, the gradient is a sum of $|\mathsf{S}|$ rank-one matrices: $\nabla_{\boldsymbol{W}_k}\mathcal{L} = \frac{1}{|\mathsf{S}|}\sum_{(\boldsymbol{x},\boldsymbol{y})\in\mathsf{S}}\nabla_{\boldsymbol{h}_k}\ell(\boldsymbol{f}(\boldsymbol{x}),\boldsymbol{y})\otimes\boldsymbol{h}_{k-1}$, where $\boldsymbol{h}_{k-1}(\boldsymbol{x})$ and $\boldsymbol{h}_k(\boldsymbol{x})$ denote the inputs and outputs of the weight matrix $\boldsymbol{W}_k$ at layer $k$, and $\otimes$ denotes the outer product. So, naïvely, one might expect the gradient $\nabla_{\boldsymbol{W}_k}\mathcal{L}$ to have a stable rank of $\min(d_k, d_{k-1}, |\mathsf{S}|)$. But it turns out to be a good assumption in practice (Yang & Hu, 2021; Yang et al., 2021; 2023). And for the definition:

**Definition 11** (Gradient summary) At a weight setting $\boldsymbol{w}$, the *gradient summary* $G$ is given by:

$$G := \frac{1}{L}\sum_{k=1}^{L}\sqrt{d_k/d_{k-1}} \cdot \|\nabla_{\boldsymbol{W}_k}\mathcal{L}(\boldsymbol{w})\|_F.$$

The gradient summary is a weighted average of gradient norms over layers. It can be thought of as a way to measure the size of the gradient while accounting for the fact that the weight matrices at different layers may be on different scales. This is related to the concept of the *gradient scale coefficient* of Philipp et al. (2017).

We now have everything we need to derive automatic gradient descent via the majorise-minimise principle:

**Theorem 2** (Automatic gradient descent) For a deep fully-connected network, under Assumptions 1 and 2 and Prescription 1, the majorisation of square loss given in Lemma 5 is minimised by setting:

$$\eta = \log \frac{1 + \sqrt{1 + 4G}}{2}, \qquad \Delta \boldsymbol{W}_k = -\frac{\eta}{L} \cdot \sqrt{d_k/d_{k-1}} \cdot \frac{\nabla_{\boldsymbol{W}_k}\mathcal{L}}{\|\nabla_{\boldsymbol{W}_k}\mathcal{L}\|_F}, \qquad \text{for all layers } k = 1, ..., L.$$

We present pseudocode for this theorem in Algorithm 1, and a PyTorch implementation in Appendix C. Via a simple derivation based on clear algorithmic principles, automatic gradient descent unifies various heuristic and theoretical ideas that have appeared in the literature:

- *Relative updates.* The update is scaled relative to the norm of the weight matrix to which it is applied—assuming the weight matrices are scaled according to Prescription 1. Such a scaling was proposed by You et al. (2017) and further explored by Carbonnelle & Vleeschouwer (2019) and Bernstein et al. (2020). There is evidence that such relative synaptic updates may occur in neuroscience (Loewenstein et al., 2011).

- *Depth scaling.* Scaling the perturbation strength like $1/L$ for networks of depth $L$ was proposed on theoretical grounds by Bernstein et al. (2020) based on analysis via deep relative trust.

- *Width scaling.* The dimensional factors of $d_k$ and $d_{k-1}$ that appear closely relate to the maximal update parameterisation of Yang & Hu (2021) designed to ensure hyperparameter transfer across network width. This connection is further explicated by Yang et al. (2023).

- *Gradient clipping.* The logarithmic dependence of the update on the gradient summary may be seen as an automatic form of *adaptive gradient clipping* (Brock et al., 2021)—a technique which clips the gradient once its magnitude surpasses a certain threshold set by a hyperparameter.

## 3.2 Convergence analysis

This section presents theoretical convergence rates for automatic gradient descent. While the spirit of the analysis is standard in optimisation theory, the details may still prove interesting for their detailed characterisation of the optimisation properties of deep networks. For instance, we propose a novel Polyak-Łojasiewicz inequality tailored to the operator structure of deep networks. We begin with two observations:

**Lemma 6** (Bounded objective) For square loss, the objective is bounded as follows:

$$\mathcal{L}(\boldsymbol{w}) \leqslant \frac{1}{|\mathsf{S}|} \sum_{(\boldsymbol{x},\boldsymbol{y}) \in \mathsf{S}} \frac{\|\boldsymbol{f}(\boldsymbol{x};\boldsymbol{w})\|_2^2 + \|\boldsymbol{y}\|_2^2}{2d_L} \leqslant 1 \text{ under Prescription 1.}$$

**Lemma 7** (Bounded gradient) For square loss, the norm of the gradient at layer $k$ is bounded as follows:

$$\|\nabla_{\boldsymbol{W}_k}\mathcal{L}\|_F \leqslant \frac{\prod_{l=1}^L \|\boldsymbol{W}_l\|_*}{\|\boldsymbol{W}_k\|_*} \cdot \sqrt{\frac{2\mathcal{L}(\boldsymbol{w})}{d_L}} \cdot \sqrt{\frac{1}{|\mathsf{S}|} \sum_{(\boldsymbol{x},\boldsymbol{y}) \in \mathsf{S}} \|\boldsymbol{x}\|_2^2} \leqslant \sqrt{2 \cdot \frac{d_{k-1}}{d_k}} \text{ under Prescription 1.}$$

These results help us prove that automatic gradient descent converges to a point where the gradient vanishes:

**Lemma 8** (Convergence rate to critical point) Consider a fully-connected network trained by automatic gradient descent (Theorem 2) and square loss for $T$ iterations. Let $G_t$ denote the gradient summary (Definition 11) at step $t \leqslant T$. Under Assumptions 1 and 2 and supposing that Prescription 1 is maintained throughout training, AGD converges at the following rate:

$$\min_{t \in \{1,...,T\}} G_t^2 \leqslant \frac{11}{T}.$$

This lemma can be converted into a convergence rate to a global minimum with one additional assumption:

**Assumption 3** (Deep Polyak-Łojasiewicz inequality) For some $\alpha > 0$, the gradient norm is lower bounded by:

$$\|\nabla_{\boldsymbol{W}_k}\mathcal{L}\|_F \geqslant \alpha \times \frac{\prod_{l=1}^{L}\|\boldsymbol{W}_l\|_*}{\|\boldsymbol{W}_k\|_*} \cdot \sqrt{\frac{2\mathcal{L}(\boldsymbol{w})}{d_L}} \cdot \sqrt{\frac{1}{|\mathsf{S}|}\sum_{(\boldsymbol{x},\boldsymbol{y})\in\mathsf{S}}\|\boldsymbol{x}\|_2^2} = \alpha \times \sqrt{2 \cdot \mathcal{L}(\boldsymbol{w}) \cdot \frac{d_{k-1}}{d_k}} \text{ under Prescription 1.}$$

This lower bound mirrors the structure of the upper bound in Lemma 7. The parameter $\alpha$ captures how much of the gradient is attenuated by small singular values in the weights and by deactivated relu units. While Polyak-Łojasiewicz inequalities are common in the literature (Liu et al., 2022), our assumption is novel in that it pays attention to the operator structure of the network. Assumption 3 leads to the following theorem:

**Theorem 3** (Convergence rate to global minima) For automatic gradient descent (Theorem 2) in the same setting as Lemma 8 but with the addition of Assumption 3, the mean squared error objective at step $T$ obeys:

$$\mathcal{L}(\boldsymbol{w}_T) \leqslant \frac{1}{\alpha^2} \times \frac{6}{T}.$$

### 3.3 Experiments

The goal of our experiments was twofold. First, we wanted to test automatic gradient descent (AGD, Algorithm 1) on a broad variety of network architectures and datasets to check that it actually works. In particular, we tested AGD on fully-connected networks (FCNs, Definition 10), and both VGG-style (Simonyan & Zisserman, 2015) and ResNet-style (He et al., 2015) convolutional neural networks on the CIFAR-10, CIFAR-100 (Krizhevsky, 2009) and ImageNet (Deng et al., 2009, ILSVRC2012) datasets with standard data augmentation. And second, to see what AGD may have to offer beyond the status quo, we wanted to compare AGD to tuned Adam and SGD baselines, as well as Adam and SGD run with their default hyperparameters.

A general theme from the experiments was that whilst AGD can train all architectures, its performance sometimes falls below that of a fully tuned baseline, both in terms of generalization and rate of convergence. While it would be nice if AGD matched the performance of fully tuned optimizers, it performed well enough to suggest that hyperparameter-free training is possible with our framework.

To get AGD working with convolutional layers, we adopted a per-submatrix normalisation scheme. Specifically, for a convolutional tensor with filters of size $\mathtt{k_x} \times \mathtt{k_y}$, we implemented the normalisation separately for each of the $\mathtt{k_x} \times \mathtt{k_y}$ submatrices of dimension $\mathtt{channels_{in}} \times \mathtt{channels_{out}}$. Since AGD does not yet support biases or affine parameters in batchnorm, we disabled these parameters in all architectures. To at least adhere to Prescription 1 at initialisation, AGD draws initial weight matrices uniform semi-orthogonal and re-scaled by a factor of $\sqrt{\mathtt{fan\_in/fan\_out}}$. Adam and SGD baselines used the PyTorch default initialisation. A PyTorch implementation of AGD reflecting these details is given in Appendix C. All experiments use square loss except ImageNet which used cross-entropy loss. Cross-entropy loss has been found to be superior to square loss for datasets with a large number of classes (Demirkaya et al., 2020; Hui & Belkin, 2021).

Our experimental results are spread across five figures, four of which appear in the appendix:

- Figure 3 presents some highlights of our results: First, AGD can train networks that Adam and SGD with default hyperparameters cannot. Second, for ResNet-18 on CIFAR-10, AGD attained performance comparable to the best-tuned performance of Adam and SGD. And third, AGD scales up to ImageNet, although the training accuracy improves slower than for SGD.

- Figure 4 displays the breadth of our experiments: from training a 16-layer fully-connected network on CIFAR-10 to training ResNet-50 on ImageNet. Adam's learning rate was tuned over the logarithmic grid $\{10^{-5}, 10^{-4}, ..., 10^{-1}\}$ while for ImageNet we used a default learning rate of 0.1 for SGD without any manual decay. AGD and Adam performed almost equally well on the depth-16, width-512 fully-connected network: 52.7% test accuracy for AGD compared to 53.5% for Adam. For ResNet-18 on CIFAR-10, Adam attained 92.9% test accuracy compared to AGD's 91.2%. On this benchmark, a fully-tuned SGD with learning rate schedule, weight decay, cross-entropy loss and bias and affine parameters can attain 93.0% test accuracy (Liu, 2017). For VGG-16 on CIFAR-100, AGD achieved 67.4% test accuracy compared to

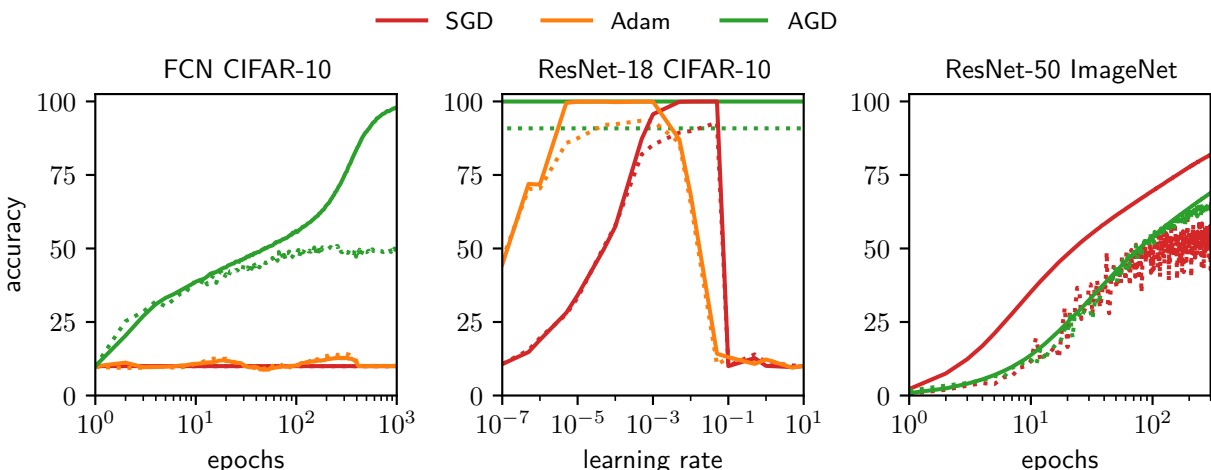

**Figure 3: Automatic gradient descent trains neural networks reliably without hyperparameters.** Solid lines show train accuracy and dotted lines show test accuracy. The networks are unregularised with biases and affine parameters disabled, as these features are not yet supported by AGD. In the **left panel**—unlike AGD—Adam and SGD failed to train a 32-layer fully-connected network on CIFAR-10 with their default learning rates of 0.001 for Adam and 0.1 for SGD. The **middle panel** displays a learning rate grid search for ResNet-18 trained on CIFAR-10. AGD attained performance comparable to the best tuned performance of Adam and SGD. In the **right panel**, AGD trained ResNet-50 on ImageNet to a top-1 test accuracy of 65.5%. The ImageNet baseline is SGD with a learning rate of 0.1 and no learning rate decay schedule.

Adam's 69.7%. Finally, on ImageNet, AGD achieved a top-1 test accuracy of 65.5% after 350 epochs. This is better than the test accuracy of SGD without weight decay or learning rate decay, but worse than "fully-loaded" SGD.

- Figure 5 compares AGD to Adam and SGD for training an eight-layer fully-connected network of width 256. Adam and SGD's learning rates were tuned over the logarithmic grid $\{10^{-5}, 10^{-4}, ..., 10^{-1}\}$. Adam's optimal learning rate of $10^{-4}$ was three orders of magnitude smaller than SGD's optimal learning rate of $10^{-1}$. SGD did not attain as low of an objective value as Adam or AGD.

- Figure 6 shows that AGD can train FCNs with widths ranging from 64 to 2048 and depths from 2 to 32 and Figure 7 shows that AGD successfully trains a four-layer FCN at varying batch size of 32 to 4096.

## 4   Discussion

This paper has proposed a new framework for deriving optimisation algorithms for non-convex composite objective functions, which are particularly prevalent in the field of machine learning and the subfield of deep learning. What we have proposed is truly a *framework*: it can be applied to a new loss function by writing down its Bregman divergence, or a new machine learning model by writing down its architectural perturbation bound. The framework is placed in the context of existing frameworks such as the majorise-minimise meta-algorithm, mirror descent and natural gradient descent.

Recent papers have proposed a paradigm of *hyperparameter transfer* where a small network is tuned and the resulting hyperparameters are transferred to a larger network (Yang et al., 2021; Bernstein, 2022). The methods and results in this paper suggest a stronger paradigm of *hyperparameter elimination*: by detailed analysis of the structure and interactions between different components of a machine learning system, we may hope—if not to outright outlaw hyperparameters—at least to reduce their abundance and opacity.

The main product of this research is automatic gradient descent (AGD), with pseudocode given in Algorithm 1 and PyTorch code given in Appendix C. The analysis leading to automatic gradient descent is elementary:

we leverage basic concepts in linear algebra such as matrix and vector norms, and use simple bounds such as the triangle inequality for vector–vector sums, and the operator norm bound for matrix–vector products. The analysis is non-asymptotic: it does not rely on taking dimensions to infinity, and deterministic: it does not involve random matrix theory. We believe that the accessibility of the analysis could make this paper a good starting point for future developments.

To validate the theory, Figure 4 demonstrates that AGD is capable of training neural networks out-of-the-box that SGD and Adam are incapable of training with default hyperparameters. While AGD trains a wide range of neural networks out of the box, more work is needed to make AGD fast and truly hyperparameter-free.

**Directions for future work**

- *Stochastic optimisation.* Automatic gradient descent is derived in the full-batch optimisation setting, but the algorithm is evaluated experimentally in the mini-batch setting. It would be interesting to try to extend our theoretical and practical methods to more faithfully address stochastic optimisation.

- *More architectures.* Automatic gradient descent is derived for fully-connected networks and extended heuristically to convolutional networks. We are curious to extend the methods to more varied architectures such as transformers (Vaswani et al., 2017) and architectural components such as biases. Since most neural networks resemble fully-connected networks in the sense that they are all just deep compound operators, we expect much of the structure of automatic gradient descent as presented to carry through.

- *Regularisation.* The present paper deals purely with the optimisation structure of deep neural networks, and little thought is given to either generalisation or regularisation. Future work could look at both theoretical and practical regularisation schemes for automatic gradient descent. It would be interesting to try to do this without introducing hyperparameters, although we suspect that when it comes to regularisation at least one hyperparameter may become necessary to tune the capacity of the network.

- *Acceleration.* We have found in some preliminary experiments that slightly increasing the update size of automatic gradient descent with a gain hyperparameter, or introducing a momentum hyperparameter, can lead to faster convergence. We emphasise that no experiment in this paper used such hyperparameters. Still, these observations may provide a valuable starting point for improving AGD in future work.

- *Operator perturbation theory.* Part of the inspiration for this paper was the idea of applying operator perturbation theory to deep learning. While perturbation theory is well-studied in the context of linear operators (Weyl, 1912; Kato, 1966; Stewart, 2006), in deep learning we are concerned with non-linear compound operators. It may be interesting to try to further extend results in perturbation theory to deep neural networks. One could imagine cataloguing the perturbation structure of different neural network building blocks, and using a result similar to deep relative trust (Lemma 4) to describe how they compound.

- *Comparison with other approaches.* Future work could carefully compare the performance of AGD to other approaches to optimisation without learning rates (Mishchenko & Defazio, 2023; Defazio & Mishchenko, 2023; Ivgi et al., 2023; Orabona & Tommasi, 2017). We do believe that these other approaches would benefit from integrating some of the ideas in this paper. For instance, in Appendix D we show that the optimal setting of the hyperparameter $d_0$ in the Prodigy optimiser (Mishchenko & Defazio, 2023) is coupled to network width.

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

# A  Further Experiments

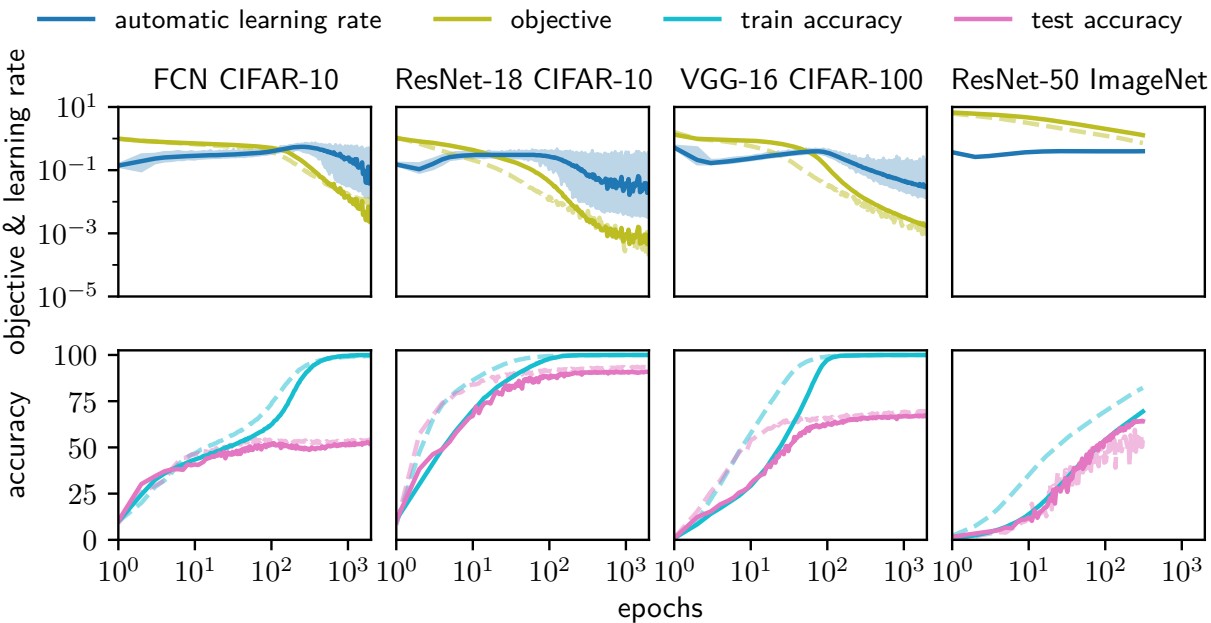

**Figure 4: Benchmarking automatic gradient descent on a range of architectures and datasets.** Solid lines are AGD and faint dashed lines are tuned Adam except for ImageNet where the dashed line is SGD with a fixed learning rate of 0.1. ImageNet used cross-entropy loss with a mini-batch size of 1024. The other experiments used square loss with a mini-batch size of 128. The **top row** plots the automatic learning rate ($\eta$ in the main text) and objective value. The maximum and minimum learning rate for each epoch is included in addition to the mean for the first three plots. The **bottom row** shows the train and test accuracy.

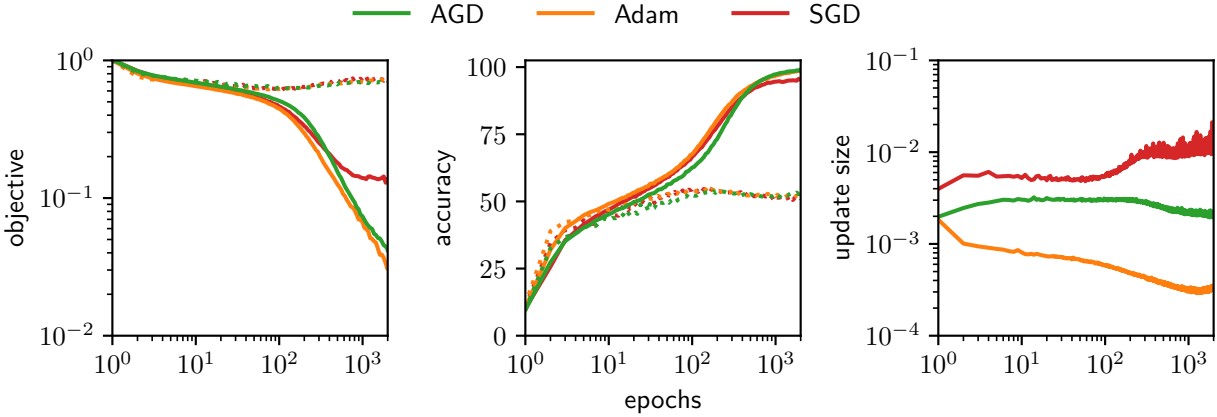

**Figure 5: Comparing automatic gradient descent to tuned Adam and SGD.** An eight-layer fully-connected network was trained on CIFAR-10 with square loss. Dotted lines show test and solid lines show train performance. The **left panel** shows the objective value: AGD and Adam attained a smaller training objective than SGD. The **middle panel** shows train and test accuracies. The **right panel** shows the relative update size averaged over layers: $\frac{1}{L}\sum_{k=1}^{L}\|\Delta\boldsymbol{W}_k\|_F/\|\boldsymbol{W}_k\|_F$. We plot the maximum, minimum and mean over an epoch.

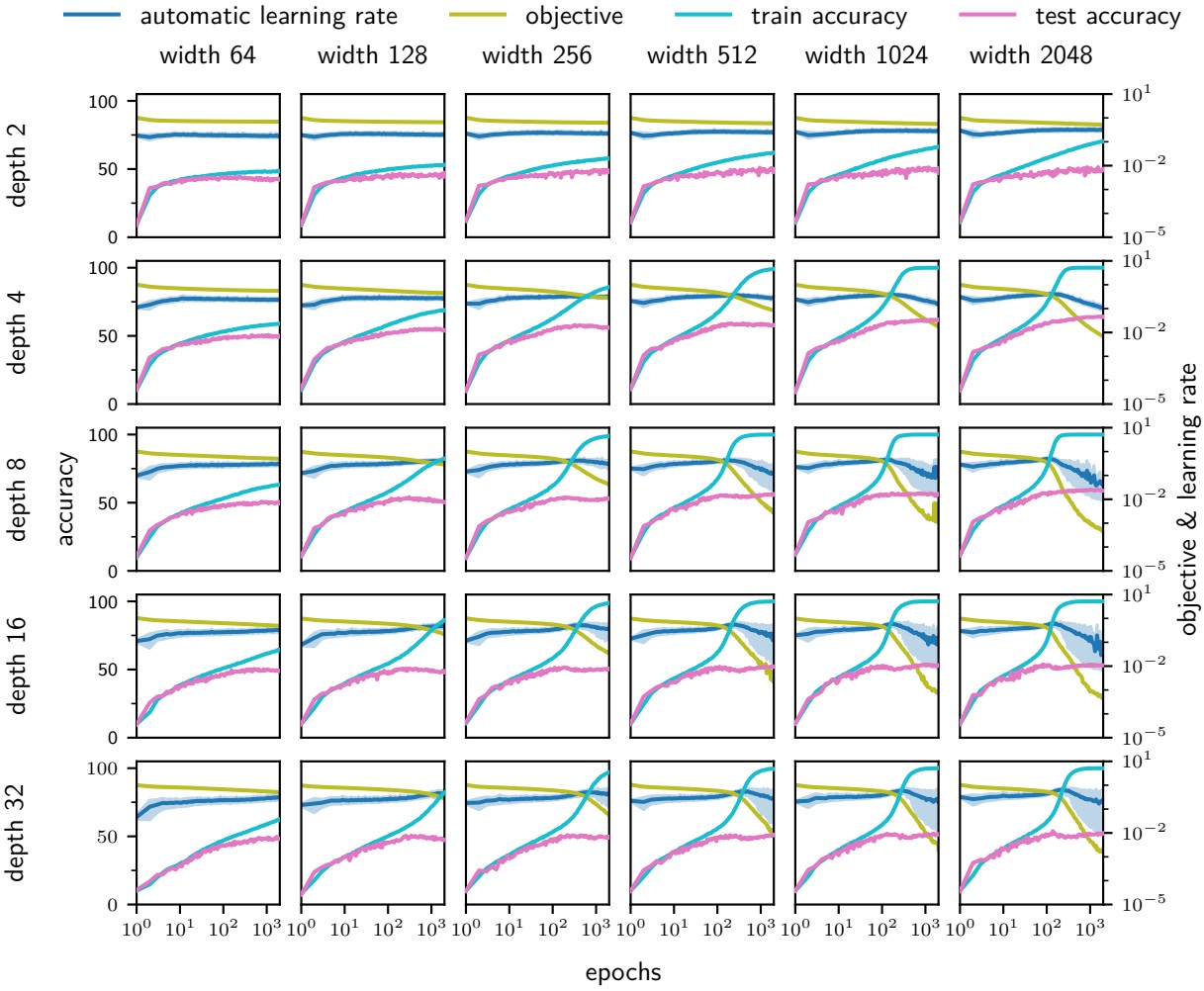

**Figure 6: Benchmarking automatic gradient descent on networks of varying width and depth.** We trained fully-connected networks on CIFAR-10 with square loss and a mini-batch size of 128. The depth ranged from 2 to 32, and the width from 64 to 2048, in powers of two. In terms of training performance, wider was always better, while depth 8 and depth 16 were superior to depth 32. In terms of test accuracy, the best performance was achieved at depth 4 and width 2048: 63.7%. The worst test performance was achieved by the smallest network of depth 2 and width 64: 42.55%. Larger networks display two broadly distinct phases of training: the automatic learning rate increases slowly while the objective decreases slowly, followed by a rapid decrease in the automatic learning rate and objective. This second phase typically coincides with reaching 100% train accuracy. See Figure 5 for a comparison between Adam, SGD and AGD for the 256-width 8-layer FCN.

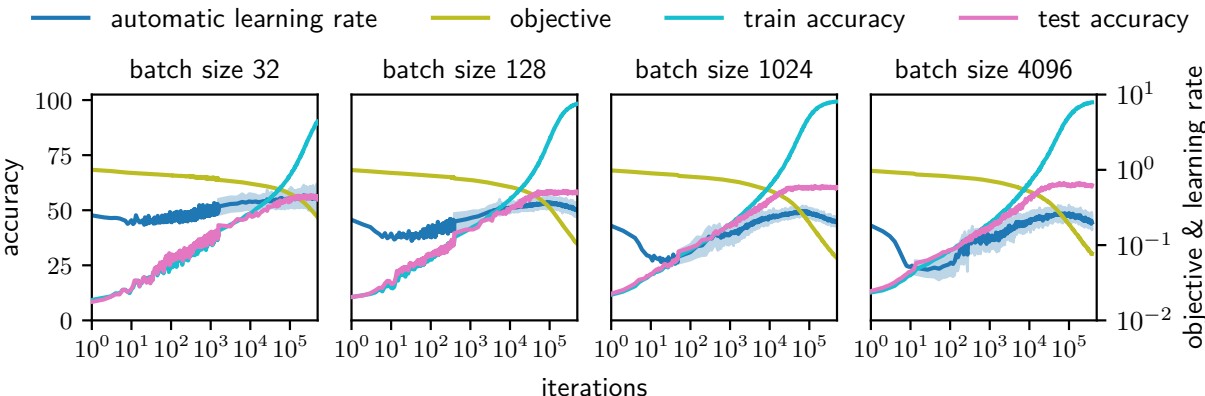

**Figure 7: Benchmarking automatic gradient descent at varying mini-batch size.** We trained four-layer fully-connected networks on CIFAR-10. The mini-batch size ranged from 32 to 4096. Test accuracy generally improved with increasing mini-batch size: the final test accuracies, in order of increasing mini-batch size, were 55.0%, 58.0%, 60.0% and 59.8%. The automatic learning rate seemed to initially dip, and this effect was more pronounced for larger mini-batch sizes. Metrics were computed every iteration during the first epoch and once per epoch from thereon—this explains the kinks visible in the plots.

# B    Proofs

Here are the proofs for the theoretical results in the main text.

**Proposition 1** (Decomposition of linearisation error) For any differentiable loss $\ell$ and any differentiable machine learning model $\boldsymbol{f}$ the linearisation error of the objective function $\mathcal{L}$ admits the following decomposition:

$$\underbrace{\Delta\mathcal{L}(\boldsymbol{w}) - \nabla_{\boldsymbol{w}}\mathcal{L}(\boldsymbol{w})^\top \Delta\boldsymbol{w}}_{\text{linearisation error of objective}} \quad = \quad \frac{1}{|\mathsf{S}|} \sum_{(\boldsymbol{x},\boldsymbol{y})\in\mathsf{S}} \nabla_{\boldsymbol{f}(\boldsymbol{x})}\ell(\boldsymbol{f}(\boldsymbol{x}),\boldsymbol{y})^\top \underbrace{\left[\Delta\boldsymbol{f}(\boldsymbol{x}) - \nabla_{\boldsymbol{w}}\boldsymbol{f}(\boldsymbol{x})\Delta\boldsymbol{w}\right]}_{\text{linearisation error of model}}$$

$$+ \frac{1}{|\mathsf{S}|} \sum_{(\boldsymbol{x},\boldsymbol{y})\in\mathsf{S}} \underbrace{\Delta\ell(\boldsymbol{f}(\boldsymbol{x}),\boldsymbol{y}) - \nabla_{\boldsymbol{f}(\boldsymbol{x})}\ell(\boldsymbol{f}(\boldsymbol{x}),\boldsymbol{y})^\top\Delta\boldsymbol{f}(\boldsymbol{x})}_{\text{linearisation error of loss}}.$$

*Proof.* By the chain rule, $\nabla_{\boldsymbol{w}}\mathcal{L}(\boldsymbol{w})^\top\Delta\boldsymbol{w} = \frac{1}{|\mathsf{S}|}\sum_{(\boldsymbol{x},\boldsymbol{y})\in\mathsf{S}}\nabla_{\boldsymbol{f}(\boldsymbol{x})}\ell(\boldsymbol{f}(\boldsymbol{x}),\boldsymbol{y})^\top\nabla_{\boldsymbol{w}}\boldsymbol{f}(\boldsymbol{x})\Delta\boldsymbol{w}$. Therefore:

$$\Delta\mathcal{L}(\boldsymbol{w}) - \nabla_{\boldsymbol{w}}\mathcal{L}(\boldsymbol{w})^\top\Delta\boldsymbol{w} = \frac{1}{|\mathsf{S}|}\sum_{(\boldsymbol{x},\boldsymbol{y})\in\mathsf{S}}\Delta\ell(\boldsymbol{f}(\boldsymbol{x}),\boldsymbol{y}) - \nabla_{\boldsymbol{f}(\boldsymbol{x})}\ell(\boldsymbol{f}(\boldsymbol{x}),\boldsymbol{y})^\top\nabla_{\boldsymbol{w}}\boldsymbol{f}(\boldsymbol{x})\Delta\boldsymbol{w}.$$

Adding and subtracting $\frac{1}{|\mathsf{S}|}\sum_{(\boldsymbol{x},\boldsymbol{y})\in\mathsf{S}}\nabla_{\boldsymbol{f}(\boldsymbol{x})}\ell(\boldsymbol{f}(\boldsymbol{x}),\boldsymbol{y})^\top\Delta\boldsymbol{f}(\boldsymbol{x})$ on the right-hand side yields the result.    □

**Lemma 1** (Bregman divergence of square loss) When $\ell$ is set to square loss, then:

$$\mathrm{bregman}_{\ell(\cdot,\boldsymbol{y})}(\boldsymbol{f}(\boldsymbol{x}),\Delta\boldsymbol{f}(\boldsymbol{x})) = \tfrac{1}{2d_L}\|\Delta\boldsymbol{f}(\boldsymbol{x})\|_2^2.$$

*Proof.* Expanding the Euclidean norms in the loss perturbation $\Delta\ell$ yields:

$$\Delta\ell(\boldsymbol{f}(\boldsymbol{x}),\boldsymbol{y}) = \tfrac{1}{2d_L}\|\boldsymbol{f}(\boldsymbol{x}) + \Delta\boldsymbol{f}(\boldsymbol{x}) - \boldsymbol{y}\|_2^2 - \tfrac{1}{2d_L}\|\boldsymbol{f}(\boldsymbol{x}) - \boldsymbol{y}\|_2^2$$
$$= \tfrac{1}{2d_L}\|\Delta\boldsymbol{f}(\boldsymbol{x})\|_2^2 + (\boldsymbol{f}(\boldsymbol{x}) - \boldsymbol{y})^\top\Delta\boldsymbol{f}(\boldsymbol{x}).$$

The result follows by identifying that $\nabla_{\boldsymbol{f}(\boldsymbol{x})}\ell(\boldsymbol{f}(\boldsymbol{x}),\boldsymbol{y})^\top\Delta\boldsymbol{f}(\boldsymbol{x}) = (\boldsymbol{f}(\boldsymbol{x}) - \boldsymbol{y})^\top\Delta\boldsymbol{f}(\boldsymbol{x})$.    □

**Lemma 2** (Bregman divergence of xent loss) When $\ell$ is set to cross-entropy loss, and if $\boldsymbol{y}^\top\mathbf{1} = 1$, then:

$$\mathrm{bregman}_{\ell(\cdot,\boldsymbol{y})}(\boldsymbol{f}(\boldsymbol{x}),\Delta\boldsymbol{f}(\boldsymbol{x})) = D_{\mathrm{KL}}\Big(\mathrm{softmax}(\boldsymbol{f}(\boldsymbol{x}))\,\Big\|\,\mathrm{softmax}(\boldsymbol{f}(\boldsymbol{x}) + \Delta\boldsymbol{f}(\boldsymbol{x}))\Big)$$
$$\leqslant \tfrac{1}{2}\|\Delta\boldsymbol{f}(\boldsymbol{x})\|_\infty^2 + \mathcal{O}(\Delta\boldsymbol{f}^3).$$

*Proof.* First, since $\sum_i \boldsymbol{y}_i = 1$, cross-entropy loss may be re-written:

$$\ell(\boldsymbol{f}(\boldsymbol{x}),\boldsymbol{y}) := -\log[\mathrm{softmax}(\boldsymbol{f}(\boldsymbol{x}))]^\top\boldsymbol{y} = -\boldsymbol{f}(\boldsymbol{x})^\top\boldsymbol{y} + \log\|\exp\boldsymbol{f}(\boldsymbol{x})\|_1.$$

The linear term $-\boldsymbol{f}(\boldsymbol{x})^\top\boldsymbol{y}$ does not contribute to the linearisation error and may be neglected. Therefore:

$$\Delta\ell(\boldsymbol{f}(\boldsymbol{x}),\boldsymbol{y}) - \nabla_{\boldsymbol{f}(\boldsymbol{x})}\ell(\boldsymbol{f}(\boldsymbol{x}),\boldsymbol{y})^\top\Delta\boldsymbol{f}(\boldsymbol{x})$$
$$= \log\|\exp(\boldsymbol{f}(\boldsymbol{x}) + \Delta\boldsymbol{f}(\boldsymbol{x}))\|_1 - \log\|\exp\boldsymbol{f}(\boldsymbol{x})\|_1 - \nabla_{\boldsymbol{f}(\boldsymbol{x})}\log\|\exp\boldsymbol{f}(\boldsymbol{x})\|_1^\top\Delta\boldsymbol{f}(\boldsymbol{x})$$
$$= \log\frac{1/\|\exp\boldsymbol{f}(\boldsymbol{x})\|_1}{1/\|\exp(\boldsymbol{f}(\boldsymbol{x}) + \Delta\boldsymbol{f}(\boldsymbol{x}))\|_1} - \frac{\exp\boldsymbol{f}(\boldsymbol{x})^\top}{\|\exp\boldsymbol{f}(\boldsymbol{x})\|_1}\Delta\boldsymbol{f}(\boldsymbol{x})$$
$$= \frac{\exp\boldsymbol{f}(\boldsymbol{x})^\top}{\|\exp\boldsymbol{f}(\boldsymbol{x})\|_1}\log\frac{\exp\boldsymbol{f}(\boldsymbol{x})/\|\exp\boldsymbol{f}(\boldsymbol{x})\|_1}{\exp(\boldsymbol{f}(\boldsymbol{x}) + \Delta\boldsymbol{f}(\boldsymbol{x}))/\|\exp(\boldsymbol{f}(\boldsymbol{x}) + \Delta\boldsymbol{f}(\boldsymbol{x}))\|_1}.$$

The final line is equivalent to $D_{\mathrm{KL}}\Big(\mathrm{softmax}(\boldsymbol{f}(\boldsymbol{x}))\,\Big\|\,\mathrm{softmax}(\boldsymbol{f}(\boldsymbol{x}) + \Delta\boldsymbol{f}(\boldsymbol{x}))\Big)$ establishing the first equality.

To establish the inequality, let $\otimes$ denote the outer product and define $p := \mathrm{softmax}(\boldsymbol{f}(\boldsymbol{x}))$. Then we have:

$$
\begin{aligned}
\Delta\ell(\boldsymbol{f}(\boldsymbol{x}), \boldsymbol{y}) - \nabla_{\boldsymbol{f}(\boldsymbol{x})}\ell(\boldsymbol{f}(\boldsymbol{x}), \boldsymbol{y})^\top \Delta\boldsymbol{f}(\boldsymbol{x}) &= \frac{1}{2}\Delta\boldsymbol{f}(\boldsymbol{x})^\top \nabla_{\boldsymbol{f}(\boldsymbol{x})}^2 \ell(\boldsymbol{f}(\boldsymbol{x}), \boldsymbol{y})\Delta\boldsymbol{f}(\boldsymbol{x}) + \mathcal{O}(\Delta\boldsymbol{f}^3) \\
&= \frac{1}{2}\Delta\boldsymbol{f}(\boldsymbol{x})^\top \nabla_{\boldsymbol{f}(\boldsymbol{x})}^2 \log\|\exp\boldsymbol{f}(\boldsymbol{x})\|_1 \Delta\boldsymbol{f}(\boldsymbol{x}) + \mathcal{O}(\Delta\boldsymbol{f}^3) \\
&= \frac{1}{2}\Delta\boldsymbol{f}(\boldsymbol{x})^\top [\mathrm{diag}(p) - p\otimes p]\Delta\boldsymbol{f}(\boldsymbol{x}) + \mathcal{O}(\Delta\boldsymbol{f}^3) \\
&\leqslant \frac{1}{2}\Delta\boldsymbol{f}(\boldsymbol{x})^\top \mathrm{diag}(p)\Delta\boldsymbol{f}(\boldsymbol{x}) + \mathcal{O}(\Delta\boldsymbol{f}^3) \\
&\leqslant \frac{1}{2}\|\Delta\boldsymbol{f}(\boldsymbol{x})\|_\infty^2 + \mathcal{O}(\Delta\boldsymbol{f}^3),
\end{aligned}
$$

where we have used that $p\otimes p$ is positive definite and then applied Hölder's inequality with $\|p\|_1 = 1$. $\qquad\square$

**Theorem 1** (Functional expansion) Consider a convex differentiable loss $\ell$ and a differentiable machine learning model $\boldsymbol{f}$. Under Assumption 1, the corresponding composite objective $\mathcal{L}$ admits the expansion:

$$
\mathcal{L}(\boldsymbol{w} + \Delta\boldsymbol{w}) = \underbrace{\mathcal{L}(\boldsymbol{w}) + \nabla_{\boldsymbol{w}}\mathcal{L}(\boldsymbol{w})^\top \Delta\boldsymbol{w}}_{\text{first-order Taylor series}} + \frac{1}{|\mathsf{S}|}\sum_{(\boldsymbol{x},\boldsymbol{y})\in\mathsf{S}} \mathrm{bregman}_{\ell(\cdot,\boldsymbol{y})}(\boldsymbol{f}(\boldsymbol{x}), \Delta\boldsymbol{f}(\boldsymbol{x})).
$$

*Proof.* The result follows by substituting Assumption 1 into Proposition 1 and applying Definition 9. $\qquad\square$

**Corollary 1** (Functional expansion of mean squared error) Under Assumption 1, for square loss:

$$
\mathcal{L}(\boldsymbol{w} + \Delta\boldsymbol{w}) = \mathcal{L}(\boldsymbol{w}) + \nabla_{\boldsymbol{w}}\mathcal{L}(\boldsymbol{w})^\top \Delta\boldsymbol{w} + \frac{1}{|\mathsf{S}|}\sum_{(\boldsymbol{x},\boldsymbol{y})\in\mathsf{S}} \frac{1}{2d_L}\|\Delta\boldsymbol{f}(\boldsymbol{x})\|_2^2.
$$

*Proof.* Combine Lemma 1 with Theorem 1 to obtain the result. $\qquad\square$

**Corollary 2** (Functional majorisation for xent loss) Under Assumption 1, for cross-entropy loss, if $\boldsymbol{y}^\top \mathbf{1} = 1$:

$$
\mathcal{L}(\boldsymbol{w} + \Delta\boldsymbol{w}) \leqslant \mathcal{L}(\boldsymbol{w}) + \nabla_{\boldsymbol{w}}\mathcal{L}(\boldsymbol{w})^\top \Delta\boldsymbol{w} + \frac{1}{|\mathsf{S}|}\sum_{(\boldsymbol{x},\boldsymbol{y})\in\mathsf{S}} \frac{1}{2}\|\Delta\boldsymbol{f}(\boldsymbol{x})\|_\infty^2 + \mathcal{O}(\Delta\boldsymbol{f}^3).
$$

*Proof.* Combine Lemma 2 with Theorem 1 to obtain the result. $\qquad\square$

**Lemma 3** (Output bound) The output norm of a fully-connected network $\boldsymbol{f}$ obeys the following bound:

$$
\|\boldsymbol{f}(\boldsymbol{x}; \boldsymbol{w})\|_2 \leqslant \left[\prod_{k=1}^{L}\|\boldsymbol{W}_k\|_*\right] \times \|\boldsymbol{x}\|_2 = \sqrt{d_L} \quad \text{under Prescription 1.}
$$

*Proof.* For any vector $\boldsymbol{v}$ and matrix $\boldsymbol{M}$ with compatible dimensions, we have that $\|\boldsymbol{M}\boldsymbol{v}\|_2 \leqslant \|\boldsymbol{M}\|_* \cdot \|\boldsymbol{v}\|_2$ and $\|\mathrm{relu}\,\boldsymbol{v}\|_2 \leqslant \|\boldsymbol{v}\|_2$. The lemma follows by applying these results recursively over the depth of the network. $\square$

**Lemma 4** (Deep relative trust) When adjusting the weights $\boldsymbol{w} = (\boldsymbol{W}_1, ..., \boldsymbol{W}_L)$ of a fully-connected network $\boldsymbol{f}$ by $\Delta\boldsymbol{w} = (\Delta\boldsymbol{W}_1, ..., \Delta\boldsymbol{W}_L)$, the induced functional perturbation $\Delta\boldsymbol{f}(\boldsymbol{x}) := \boldsymbol{f}(\boldsymbol{x}; \boldsymbol{w} + \Delta\boldsymbol{w}) - \boldsymbol{f}(\boldsymbol{x}; \boldsymbol{w})$ obeys:

$$
\|\Delta\boldsymbol{f}(\boldsymbol{x})\|_2 \leqslant \left[\prod_{k=1}^{L}\|\boldsymbol{W}_k\|_*\right] \times \|\boldsymbol{x}\|_2 \times \left[\prod_{k=1}^{L}\left(1 + \frac{\|\Delta\boldsymbol{W}_k\|_*}{\|\boldsymbol{W}_k\|_*}\right) - 1\right] \leqslant \sqrt{d_L} \times (\exp\eta - 1) \quad \text{under Prescription 1.}
$$

*Proof.* We proceed by induction. First, consider a network with $L = 1$ layers: $\boldsymbol{f}(\boldsymbol{x}) = \boldsymbol{W}_1\boldsymbol{x}$. Observe that $\|\Delta\boldsymbol{f}(\boldsymbol{x})\|_2 = \|\Delta\boldsymbol{W}_1\boldsymbol{x}\|_2 \leqslant \|\Delta\boldsymbol{W}_1\|_* \cdot \|\boldsymbol{x}\|_2$ as required. Next, assume that the result holds for a network $\boldsymbol{g}(\boldsymbol{x})$ with $L - 1$ layers and consider adding a layer to obtain $\boldsymbol{f}(\boldsymbol{x}) = \boldsymbol{W}_L \circ \mathrm{relu} \circ \boldsymbol{g}(\boldsymbol{x})$. Then:

$$
\begin{aligned}
\|\Delta\boldsymbol{f}(\boldsymbol{x})\|_2 &= \|(\boldsymbol{W}_L + \Delta\boldsymbol{W}_L)\circ\mathrm{relu}\circ(\boldsymbol{g}(\boldsymbol{x}) + \Delta\boldsymbol{g}(\boldsymbol{x})) - \boldsymbol{W}_L \circ\mathrm{relu}\circ\boldsymbol{g}(\boldsymbol{x})\|_2 \\
&= \|\boldsymbol{W}_L(\mathrm{relu}\circ(\boldsymbol{g}(\boldsymbol{x}) + \Delta\boldsymbol{g}(\boldsymbol{x})) - \mathrm{relu}\circ\boldsymbol{g}(\boldsymbol{x})) + \Delta\boldsymbol{W}_L(\mathrm{relu}\circ(\boldsymbol{g}(\boldsymbol{x}) + \Delta\boldsymbol{g}(\boldsymbol{x})) - \mathrm{relu}(0))\|_2 \\
&\leqslant \|\boldsymbol{W}_L\|_* \cdot \|\Delta\boldsymbol{g}(\boldsymbol{x})\|_2 + \|\Delta\boldsymbol{W}_L\|_* \cdot (\|\boldsymbol{g}(\boldsymbol{x})\|_2 + \|\Delta\boldsymbol{g}(\boldsymbol{x})\|_2) \\
&= (\|\boldsymbol{W}_L\|_* + \|\Delta\boldsymbol{W}_L\|_*)\cdot\|\Delta\boldsymbol{g}(\boldsymbol{x})\|_2 + \|\Delta\boldsymbol{W}_L\|_* \cdot \|\boldsymbol{g}(\boldsymbol{x})\|_2,
\end{aligned}
$$

where the inequality follows by applying the triangle inequality, the operator norm bound, the fact that relu is one-Lipschitz, and a further application of the triangle inequality. But by the inductive hypothesis and Lemma 3, the right-hand side is bounded by:

$$(\|\boldsymbol{W}_L\|_* + \|\Delta\boldsymbol{W}_L\|_*)\left[\prod_{k=1}^{L-1}\left(1 + \frac{\|\Delta\boldsymbol{W}_k\|_*}{\|\boldsymbol{W}_k\|_*}\right) - 1\right] \times \left[\prod_{k=1}^{L-1}\|\boldsymbol{W}_k\|_*\right] \times \|\boldsymbol{x}\|_2 + \|\Delta\boldsymbol{W}_L\|_* \times \left[\prod_{k=1}^{L-1}\|\boldsymbol{W}_k\|_*\right] \times \|\boldsymbol{x}\|_2$$

$$= \left[\prod_{k=1}^{L}\left(1 + \frac{\|\Delta\boldsymbol{W}_k\|_*}{\|\boldsymbol{W}_k\|_*}\right) - 1\right] \times \left[\prod_{k=1}^{L}\|\boldsymbol{W}_k\|_*\right] \times \|\boldsymbol{x}\|_2.$$

The induction is complete. To further bound this result under Prescription 1, observe that the product $\left[\prod_{k=1}^{L}\|\boldsymbol{W}_k\|_*\right] \times \|\boldsymbol{x}\|_2$ telescopes to just $\sqrt{d_L}$, while the other product satisfies:

$$\left[\prod_{k=1}^{L}\left(1 + \frac{\|\Delta\boldsymbol{W}_k\|_*}{\|\boldsymbol{W}_k\|_*}\right) - 1\right] = \left(1 + \frac{\eta}{L}\right)^L - 1 \leqslant \lim_{L\to\infty}\left(1 + \frac{\eta}{L}\right)^L - 1 = \exp\eta - 1.$$

Combining these observations yields the result. □

**Lemma 5** (Exponential majorisation) For an FCN with square loss, under Assumption 1 and Prescription 1:

$$\mathcal{L}(\boldsymbol{w} + \Delta\boldsymbol{w}) \leqslant \mathcal{L}(\boldsymbol{w}) + \frac{\eta}{L}\sum_{k=1}^{L}\left[\sqrt{d_k/d_{k-1}} \times \text{tr}\frac{\Delta\boldsymbol{W}_k^\top\nabla_{\boldsymbol{W}_k}\mathcal{L}}{\|\Delta\boldsymbol{W}_k\|_*}\right] + \tfrac{1}{2}\left(\exp\eta - 1\right)^2.$$

*Proof.* Substitute Lemma 4 into Corollary 1 and decompose $\nabla_{\boldsymbol{w}}\mathcal{L}(\boldsymbol{w})^\top\Delta\boldsymbol{w} = \sum_{k=1}^{L}\text{tr}(\Delta\boldsymbol{W}_k^\top\nabla_{\boldsymbol{W}_k}\mathcal{L})$. The result follows by realising that under Prescription 1, the perturbations satisfy $\|\Delta\boldsymbol{W}_k\|_* = \sqrt{d_k/d_{k-1}} \cdot \frac{\eta}{L}$. □

**Theorem 2** (Automatic gradient descent) For a deep fully-connected network, under Assumptions 1 and 2 and Prescription 1, the majorisation of square loss given in Lemma 5 is minimised by setting:

$$\eta = \log\frac{1 + \sqrt{1 + 4G}}{2}, \qquad \Delta\boldsymbol{W}_k = -\frac{\eta}{L}\cdot\sqrt{d_k/d_{k-1}}\cdot\frac{\nabla_{\boldsymbol{W}_k}\mathcal{L}}{\|\nabla_{\boldsymbol{W}_k}\mathcal{L}\|_F}, \qquad \text{for all layers } k = 1,...,L.$$

*Proof.* The inner product $\text{tr}\frac{\Delta\boldsymbol{W}_k^\top\nabla_{\boldsymbol{W}_k}\mathcal{L}}{\|\Delta\boldsymbol{W}_k\|_*}$ that appears in Lemma 5 is most negative when the perturbation $\Delta\boldsymbol{W}_k$ satisfies $\Delta\boldsymbol{W}_k/\|\Delta\boldsymbol{W}_k\|_* = -\nabla_{\boldsymbol{W}_k}\mathcal{L}/\|\nabla_{\boldsymbol{W}_k}\mathcal{L}\|_*$. Substituting this result back into Lemma 5 yields:

$$\mathcal{L}(\boldsymbol{w} + \Delta\boldsymbol{w}) \leqslant \mathcal{L}(\boldsymbol{w}) - \frac{\eta}{L}\sum_{k=1}^{L}\left[\sqrt{d_k/d_{k-1}} \times \frac{\|\nabla_{\boldsymbol{W}_k}\mathcal{L}\|_F^2}{\|\nabla_{\boldsymbol{W}_k}\mathcal{L}\|_*}\right] + \tfrac{1}{2}\left(\exp\eta - 1\right)^2.$$

Under Assumption 2, we have that $\|\nabla_{\boldsymbol{W}_k}\mathcal{L}\|_F^2/\|\nabla_{\boldsymbol{W}_k}\mathcal{L}\|_* = \|\nabla_{\boldsymbol{W}_k}\mathcal{L}\|_F$ and so this inequality simplifies to:

$$\mathcal{L}(\boldsymbol{w} + \Delta\boldsymbol{w}) \leqslant \mathcal{L}(\boldsymbol{w}) - \eta \cdot G + \tfrac{1}{2}\left(\exp\eta - 1\right)^2.$$

Taking the derivative of the right-hand side with respect to $\eta$ and setting it to zero yields $(\exp\eta - 1)\exp\eta = G$. Applying the quadratic formula and retaining the positive solution yields $\exp\eta = \frac{1}{2}(1 + \sqrt{1 + 4G})$. Combining this with the relation that $\Delta\boldsymbol{W}_k/\|\Delta\boldsymbol{W}_k\|_* = -\nabla_{\boldsymbol{W}_k}\mathcal{L}/\|\nabla_{\boldsymbol{W}_k}\mathcal{L}\|_*$ and applying that $\|\Delta\boldsymbol{W}_k\|_* = \sqrt{d_k/d_{k-1}} \cdot \frac{\eta}{L}$ by Prescription 1 yields the result. □

**Lemma 6** (Bounded objective) For square loss, the objective is bounded as follows:

$$\mathcal{L}(\boldsymbol{w}) \leqslant \frac{1}{|\mathsf{S}|}\sum_{(\boldsymbol{x},\boldsymbol{y})\in\mathsf{S}}\frac{\|\boldsymbol{f}(\boldsymbol{x};\boldsymbol{w})\|_2^2 + \|\boldsymbol{y}\|_2^2}{2d_L} \leqslant 1 \text{ under Prescription 1}.$$

*Proof.* The result follows by the following chain of inequalities:

$$\mathcal{L}(\boldsymbol{w}) := \frac{1}{|\mathsf{S}|}\sum_{(\boldsymbol{x},\boldsymbol{y})\in\mathsf{S}}\frac{1}{2d_L}\|\boldsymbol{f}(\boldsymbol{x};\boldsymbol{w}) - \boldsymbol{y}\|_2^2 \leqslant \frac{1}{|\mathsf{S}|}\sum_{(\boldsymbol{x},\boldsymbol{y})\in\mathsf{S}}\frac{1}{2d_L}(\|\boldsymbol{f}(\boldsymbol{x};\boldsymbol{w})\|_2^2 + \|\boldsymbol{y}\|_2^2) \leqslant \frac{1}{|\mathsf{S}|}\sum_{(\boldsymbol{x},\boldsymbol{y})\in\mathsf{S}}\frac{d_L + d_L}{2d_L} = 1,$$

where the second inequality holds under Prescription 1. □

**Lemma 7** (Bounded gradient) For square loss, the norm of the gradient at layer $k$ is bounded as follows:

$$\|\nabla_{\boldsymbol{W}_k}\mathcal{L}\|_F \leqslant \frac{\prod_{l=1}^{L}\|\boldsymbol{W}_l\|_*}{\|\boldsymbol{W}_k\|_*} \cdot \sqrt{\frac{2\mathcal{L}(\boldsymbol{w})}{d_L}} \cdot \sqrt{\frac{1}{|\mathsf{S}|}\sum_{(\boldsymbol{x},\boldsymbol{y})\in\mathsf{S}}\|\boldsymbol{x}\|_2^2} \leqslant \sqrt{2 \cdot \frac{d_{k-1}}{d_k}} \quad \text{under Prescription 1.}$$

*Proof.* By the chain rule, the gradient of mean square error objective may be written:

$$\nabla_{\boldsymbol{W}_k}\mathcal{L}(\boldsymbol{w}) = \frac{1}{|\mathsf{S}|}\sum_{(\boldsymbol{x},\boldsymbol{y})\in\mathsf{S}}\frac{1}{d_L}(\boldsymbol{f}(\boldsymbol{x};\boldsymbol{w}) - \boldsymbol{y})^\top \boldsymbol{W}_L \cdot \boldsymbol{D}_{L-1}\boldsymbol{W}_{L-1}\ldots\boldsymbol{D}_{k+1}\boldsymbol{W}_{k+1} \cdot \boldsymbol{D}_k \otimes \boldsymbol{D}_{k-1}\boldsymbol{W}_{k-1}\ldots\boldsymbol{D}_1\boldsymbol{W}_1\boldsymbol{x},$$

where $\otimes$ denotes the outer product and $\boldsymbol{D}_k$ denotes a diagonal matrix whose entries are one when relu is active and zero when relu is inactive. Since the operator norm $\|\boldsymbol{D}_k\|_* = 1$, we have that the Frobenius norm $\|\nabla_{\boldsymbol{W}_k}\mathcal{L}(\boldsymbol{w})\|_F$ is bounded from above by:

$$\frac{1}{|\mathsf{S}|}\sum_{(\boldsymbol{x},\boldsymbol{y})\in\mathsf{S}}\frac{1}{d_L}\|(\boldsymbol{f}(\boldsymbol{x};\boldsymbol{w}) - \boldsymbol{y})^\top \boldsymbol{W}_L \cdot \boldsymbol{D}_{L-1}\boldsymbol{W}_{L-1}\ldots\boldsymbol{D}_{k+1}\boldsymbol{W}_{k+1} \cdot \boldsymbol{D}_k \otimes \boldsymbol{D}_{k-1}\boldsymbol{W}_{k-1}\ldots\boldsymbol{D}_1\boldsymbol{W}_1\boldsymbol{x}\|_F$$

$$= \frac{1}{|\mathsf{S}|}\sum_{(\boldsymbol{x},\boldsymbol{y})\in\mathsf{S}}\frac{1}{d_L}\|(\boldsymbol{f}(\boldsymbol{x};\boldsymbol{w}) - \boldsymbol{y})^\top \boldsymbol{W}_L \cdot \boldsymbol{D}_{L-1}\boldsymbol{W}_{L-1}\ldots\boldsymbol{D}_{k+1}\boldsymbol{W}_{k+1} \cdot \boldsymbol{D}_k\|_2 \cdot \|\boldsymbol{D}_{k-1}\boldsymbol{W}_{k-1}\ldots\boldsymbol{D}_1\boldsymbol{W}_1\boldsymbol{x}\|_2$$

$$\leqslant \frac{1}{|\mathsf{S}|}\sum_{(\boldsymbol{x},\boldsymbol{y})\in\mathsf{S}}\frac{1}{d_L}\|\boldsymbol{f}(\boldsymbol{x};\boldsymbol{w}) - \boldsymbol{y}\|_2 \cdot \|\boldsymbol{W}_L\|_* \cdot \|\boldsymbol{W}_{L-1}\|_* \ldots \|\boldsymbol{W}_{k+1}\|_* \cdot \|\boldsymbol{W}_{k-1}\|_* \ldots \|\boldsymbol{W}_1\|_* \cdot \|\boldsymbol{x}\|_2$$

$$= \frac{\prod_{l=1}^{L}\|\boldsymbol{W}_l\|_*}{\|\boldsymbol{W}_k\|} \times \frac{1}{|\mathsf{S}|}\sum_{(\boldsymbol{x},\boldsymbol{y})\in\mathsf{S}}\frac{1}{d_L}\|\boldsymbol{f}(\boldsymbol{x};\boldsymbol{w}) - \boldsymbol{y}\|_2 \cdot \|\boldsymbol{x}\|_2$$

$$\leqslant \frac{\prod_{l=1}^{L}\|\boldsymbol{W}_l\|_*}{\|\boldsymbol{W}_k\|_*} \cdot \frac{1}{\sqrt{d_L}}\sqrt{\frac{2}{|\mathsf{S}|}\sum_{(\boldsymbol{x},\boldsymbol{y})\in\mathsf{S}}\frac{1}{2d_L}\|\boldsymbol{f}(\boldsymbol{x};\boldsymbol{w}) - \boldsymbol{y}\|_2^2} \cdot \sqrt{\frac{1}{|\mathsf{S}|}\sum_{(\boldsymbol{x},\boldsymbol{y})\in\mathsf{S}}\|\boldsymbol{x}\|_2^2}$$

$$= \frac{\prod_{l=1}^{L}\|\boldsymbol{W}_l\|_*}{\|\boldsymbol{W}_k\|_*} \cdot \sqrt{\frac{2\mathcal{L}(\boldsymbol{w})}{d_L}} \cdot \sqrt{\frac{1}{|\mathsf{S}|}\sum_{(\boldsymbol{x},\boldsymbol{y})\in\mathsf{S}}\|\boldsymbol{x}\|_2^2}.$$

In the above argument, the first inequality follows by recursive application of the operator norm upper bound, and the second inequality follows from the Cauchy-Schwarz inequality. The right-hand side simplifies under Prescription 1, and we may apply Lemma 6 to obtain:

$$\|\nabla_{\boldsymbol{W}_k}\mathcal{L}(\boldsymbol{w})\|_F \leqslant \frac{\prod_{l=1}^{L}\|\boldsymbol{W}_l\|_*}{\|\boldsymbol{W}_k\|_*} \cdot \sqrt{\frac{2\mathcal{L}(\boldsymbol{w})}{d_L}} \cdot \sqrt{\frac{1}{|\mathsf{S}|}\sum_{(\boldsymbol{x},\boldsymbol{y})\in\mathsf{S}}\|\boldsymbol{x}\|_2^2} \leqslant \frac{\sqrt{d_L/d_0}}{\sqrt{d_k/d_{k-1}}} \cdot \sqrt{\frac{2}{d_L}} \cdot \sqrt{d_0} = \sqrt{2} \cdot \sqrt{\frac{d_{k-1}}{d_k}}.$$

**Lemma 8** (Convergence rate to critical point) Consider a fully-connected network trained by automatic gradient descent (Theorem 2) and square loss for $T$ iterations. Let $G_t$ denote the gradient summary (Definition 11) at step $t \leqslant T$. Under Assumptions 1 and 2 and supposing that Prescription 1 is maintained throughout training, AGD converges at the following rate:

$$\min_{t\in\{1,\ldots,T\}} G_t^2 \leqslant \frac{11}{T}.$$

*Proof.* Theorem 2 prescribes that $\exp\eta = \frac{1}{2}(1 + \sqrt{1 + 4G})$, and so $\eta = \log\left(1 + \frac{\sqrt{1+4G}-1}{2}\right)$. We begin by proving some useful auxiliary bounds. By Lemma 7 and Prescription 1, the gradient summary is bounded by:

$$G := \frac{1}{L}\sum_{k=1}^{L}\sqrt{d_k/d_{k-1}} \cdot \|\nabla_{\boldsymbol{W}_k}\mathcal{L}(\boldsymbol{w})\|_F \leqslant \frac{1}{L}\sum_{k=1}^{L}\sqrt{2} < 2.$$

The fact that the gradient summary $G$ is less than two is important because, for $x \leqslant 1$, we have that $\log(1 + x) \geqslant x\log 2$. In turn, this implies that since $G < 2$, we have that $\eta = \log\frac{1+\sqrt{1+4G}}{2} \geqslant \frac{\sqrt{1+4G}-1}{2}\log 2$. It will also be important to know that for $G < 2$, we have that $\frac{1}{2} \cdot G \leqslant \frac{\sqrt{1+4G}-1}{2} \leqslant G$.

With these bounds in hand, the analysis becomes fairly standard. By an intermediate step in the proof of Theorem 2, the change in objective across a single step is bounded by:

$$
\begin{aligned}
\mathcal{L}(\boldsymbol{w} + \Delta\boldsymbol{w}) - \mathcal{L}(\boldsymbol{w}) &\leqslant -\eta \cdot G + \tfrac{1}{2}\left(\exp\eta - 1\right)^2 \\
&\leqslant -\tfrac{\sqrt{1+4G}-1}{2}\left(G\log 2 - \tfrac{1}{2}\tfrac{\sqrt{1+4G}-1}{2}\right) \\
&\leqslant -\tfrac{1}{2}\cdot\left(\log 2 - \tfrac{1}{2}\right)\cdot G^2 \leqslant -G^2/11,
\end{aligned}
$$

where the second and third inequalities follow by our auxiliary bounds. Letting $G_t$ denote the gradient summary at step $t$, averaging this bound over time steps and applying the telescoping property yields:

$$
\min_{t\in[1,\ldots,T]} G_t^2 \leqslant \frac{1}{T}\sum_{t=1}^{T} G_t^2 \leqslant \frac{11}{T}\sum_{t=1}^{T}\mathcal{L}(\boldsymbol{w}_t) - \mathcal{L}(\boldsymbol{w}_{t+1}) = \frac{11}{T}\cdot\left(\mathcal{L}(\boldsymbol{w}_1) - \mathcal{L}(\boldsymbol{w}_T)\right) \leqslant \frac{11}{T},
$$

where the final inequality follows by Lemma 6 and the fact that $\mathcal{L}(\boldsymbol{w}_T) \geqslant 0$.

**Theorem 3** (Convergence rate to global minima) For automatic gradient descent (Theorem 2) in the same setting as Lemma 8 but with the addition of Assumption 3, the mean squared error objective at step $T$ obeys:

$$
\mathcal{L}(\boldsymbol{w}_T) \leqslant \frac{1}{\alpha^2}\times\frac{6}{T}.
$$

*Proof.* By Assumption 3, the gradient summary at time step $t$ must satisfy $G_t \geqslant \alpha \times \sqrt{2\cdot\mathcal{L}(\boldsymbol{w}_t)}$. Therefore the objective at time step $t$ is bounded by $\mathcal{L}(\boldsymbol{w}_t) \leqslant G_t^2/(2\alpha^2)$. Combining with Lemma 8 then yields that:

$$
\mathcal{L}(\boldsymbol{w}_T) = \min_{t\in[1,\ldots,T]}\mathcal{L}(\boldsymbol{w}_t) \leqslant \frac{1}{2\alpha^2}\min_{t\in[1,\ldots,T]} G_t^2 \leqslant \frac{6}{\alpha^2 T}.
$$

The proof is complete. $\qquad\square$

## C   PyTorch Implementation

The following code implements automatic gradient descent in PyTorch (Paszke et al., 2019). We include a single gain hyperparameter which controls the update size and may be increased from its default value of 1.0 to slightly accelerate training. We emphasise that all the results reported in the paper used a gain of unity.

```python
import math
import torch

from torch.nn.init import orthogonal_

def singular_value(p):
    sv = math.sqrt(p.shape[0] / p.shape[1])
    if p.dim() == 4:
        sv /= math.sqrt(p.shape[2] * p.shape[3])
    return sv

class AGD:
    @torch.no_grad()
    def __init__(self, net, gain=1.0):

        self.net = net
        self.depth = len(list(net.parameters()))
        self.gain = gain

        for p in net.parameters():
            if p.dim() == 1: raise Exception("Biases are not supported.")
            if p.dim() == 2: orthogonal_(p)
            if p.dim() == 4:
                for kx in range(p.shape[2]):
                    for ky in range(p.shape[3]):
                        orthogonal_(p[:,:,kx,ky])
            p *= singular_value(p)

    @torch.no_grad()
    def step(self):

        G = 0
        for p in self.net.parameters():
            G += singular_value(p) * p.grad.norm(dim=(0,1)).sum()
        G /= self.depth

        log = math.log(0.5 * (1 + math.sqrt(1 + 4*G)))

        for p in self.net.parameters():
            factor = singular_value(p) / p.grad.norm(dim=(0,1), keepdim=True)
            p -= self.gain * log / self.depth * factor * p.grad
```

# D  Other optimisers

| Algorithm | Reference | Number of Hyperparameters |
|---|---|---|
| SGD+Armijo | Vaswani et al. (2019) | 5 |
| Parabolic Line Search | Mutschler & Zell (2020) | 1 |
| L4 | Rolinek & Martius (2018) | 5 |
| HyperSGD | Chandra et al. (2022) | many |
| D-Adaptation | Defazio & Mishchenko (2023) | 1 |
| DoG | Ivgi et al. (2023) | 2+ |
| Stochastic Polyak step-size | Loizou et al. (2021) | 2 |
| COCOB | Orabona & Tommasi (2017) | 1 |
| Prodigy | Mishchenko & Defazio (2023) | 2+ |

**Table 2: Other approaches to optimisation without learning rates.** This table lists some optimisers and the number of hyperparameters each one uses (not including optional ones like learning rate decay, or other training hyperparameters like batch size).

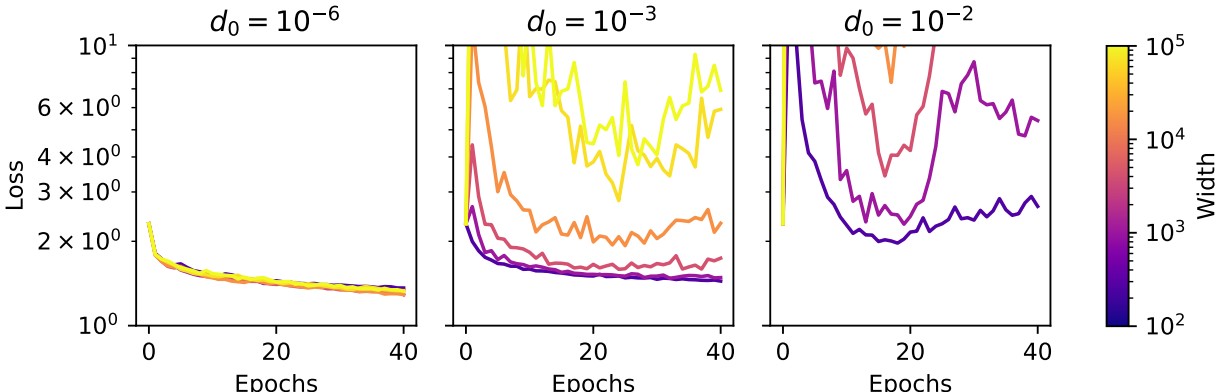

**Figure 8: The required initial step size $d_0$ in Prodigy is coupled to width.** We trained depth-2 fully connected networks with different widths on CIFAR-10 with the Prodigy optimizer (Mishchenko & Defazio, 2023). We varied the $d_0$ hyperparameter and for each setting of $d_0$ we trained networks of varying width. The plots demonstrate that the optimal setting of $d_0$ in Prodigy is coupled to width in an undesirable way.

