# OpenReview forum: "Automatic Gradient Descent: Toward Deep Learning without Hyperparameters"
_TMLR — Rejected by TMLR_

### Review · Reviewer_6E7M · 2024-11-04

**Summary Of Contributions:**

This paper introduces Automatic Gradient Descent (AGD), an optimizer for deep learning that eliminates the need for traditional hyperparameters like learning rate and initialization variance. AGD follows three main steps: functional expansion, architectural perturbation bounds, and majorise-minimise. Under three questionable assumptions, AGD is theoretically guaranteed to converge to the global minimizer.

Empirical validation demonstrates AGD’s effectiveness across diverse architectures, performing competitively with tuned optimizers without manual hyperparameter adjustments. This framework signals a shift towards hyperparameter-free optimization, reducing tuning costs and addressing reproducibility issues.

**Audience:**

Yes

**Broader Impact Concerns:**

This is a theoretical work. There is no such concern.

**Claims And Evidence:**

Yes

**Requested Changes:**

Please see weaknesses and address the raised concerns.

Proof of Theorem 2: In the first sentence, "that appears in Lemma 5 is most negative," delete the word "most."

**Strengths And Weaknesses:**

Strengths:

Impressive Theoretical Contribution: The paper demonstrates global convergence of Automatic Gradient Descent (AGD), a significant achievement given that global convergence for gradient descent (GD) has not been completely solved even for two-layer networks.

Hyperparameter-Free Optimization: AGD eliminates traditional hyperparameters like learning rate, simplifying training and reducing computational costs, which could improve reproducibility.

Clear Derivation: The step-by-step development of AGD, covering functional expansion, architectural perturbation bounds, and majorise-minimise, is well-presented and accessible.

Weaknesses: The primary issues stem from the three questionable assumptions. Firstly, for a non-trivial network, it is nearly impossible for any one of Assumptions 1-3 to hold exactly for all weight matrices $w$. However, from the proof, it appears that it only requires Assumptions 1-3 to hold at initialization and to hold throughout the iterations.

Starting with the least questionable assumption, Assumption 1, I will refer to the "orthogonality of model linearization error" as Property 1 rather than an assumption. If we assume that Property 1 holds at initialization, the authors need to explain why this property would remain unchanged, or nearly so, throughout the iterations. For example, the infinite-width limit approach in the neural tangent kernel (NTK) regime might be necessary here. If the authors can at least demonstrate Property 1 for a general deep network without requiring very large width, along with a mathematical justification, it would then be reasonable to elevate this to Assumption 1 for a broad class of neural networks.

Similarly, I will refer to Assumption 2 in this paper as Property 2. It seems impossible for Property 2 to hold exactly during the iterations, since $rank_{stable}A=1$ if and only if $rank(A)=1$, and $A$ here is the sum of $|S|$ rank-one matrices. It also seems unlikely for Property 2 to hold approximately during iterations, since $rank_{stable}A=1+o(1)$ if and only if the largest singular value satisfies $\sigma^2_1(A)>>\sum_{i\geq 2} \sigma^2_i(A)$. The authors need to justify that there exists a class of networks for which Property 2 holds approximately throughout the iterations. Simply citing other papers seems insufficient.

While Assumption 3 is common in the literature, it directly contradicts the non-convex nature of the loss surface in deep neural networks. For a non-convex loss surface, there exist stationary points that are not global minimizers, denoted as $W^*$. Thus, at least at $W^*$, Assumption 3 does not hold. If Assumption 3 is meant to hold only along the entire training trajectory, it would imply that the algorithm is able to avoid all existing stationary points that are not global minimizers. It seems almost too good to be true that a gradient-based method could inherently avoid saddles and local minima. The neural tangent kernel (NTK) regime might be necessary here.

It appears that the NTK regime could address the issues with Assumptions 1 and 3, though it may not resolve the challenges posed by Assumption 2.

---

> ### Author Response · Authors · 2025-01-02
>
> Dear Reviewer 6E7M, thank you for your thorough review. We are glad to hear that you found our derivation to be clear and our theoretical contribution to be impressive.
>
> In response to your general comments about the assumptions, we want to emphasise that we are not claiming that the assumptions exactly hold in practice. The goal of our paper is to produce a formal procedure for optimizer design. What we are showing is that the overall framework combined with the specific assumptions allows us to derive an optimizer that is explicitly aware of neural architecture. Each assumption highlights a central quantity in the theory and we invite future work to consider relaxing the assumptions or replacing them with alternatives. We do see our approach as a step forward in contrast to contemporaneous work that makes much stronger assumptions: ignoring the structure of the neural network and studying convex optimization, or assuming that weight updates to different network layers do not interact, for two common examples.
>
> Here we provide more detailed discussion of each assumption, as well as possible alternative strategies that future work could explore. We propose updating the paper to include this additional discussion.
>
> **Assumption 1:** This deals with the fact that a smoothness model or a majorisation of a deep learning loss function must deal with the linearisation error of the network by some means. Here, for simplicity, we assume that this term can be ignored. But another approach would be to bound the linearisation error of the network. Such bounds are given for deep linear networks in Lemma 6.2 of [this PhD thesis](https://arxiv.org/abs/2210.10101). It is hard to extend this approach to ReLU networks because we want an O(∆w^2) bound but ReLU is non-smooth so will not entertain such a bound. However, it should be possible to extend this approach to networks built out of smooth components such as GELU and SwiGLU nonlinearities, which are becoming popular nowadays.
>
> **Prescription 1:** This deals with the fact that, without some control on various norms of quantities inside the network (such as weights and inputs), a network will not exhibit a global majorization. For example, as the norm of a weight matrix grows, the network becomes increasingly sensitive to perturbations to the other layers. In this paper, we deal with this issue by assuming various quantities are of known and fixed size. This could inspire future work to consider projected gradient descent algorithms or regularization strategies designed to maintain these properties.
>
> **Assumption 2:** This assumption deals with basic conditioning properties of the gradient. While [other work has presented evidence for this assumption](https://arxiv.org/abs/2310.17813), the assumption could easily be removed and we would instead see the spectral norm instead of the Frobenius norm appearing in the analysis. In fact, this observation is currently inspiring interest in optimization algorithms for deep networks grounded in spectral norm geometry.
>
> **Assumption 3:** This assumption is mentioned simply to illustrate a setting where convergence to critical points implies convergence to global minima. We are not suggesting that this assumption actually holds in practice. The point is to write down a Polyak-Łojasiewicz-style inequality that reflects the actual layered structure of the neural network. This invites the reader to think about how neighbouring layers need to interact for such an inequality to hold. To the best of our knowledge, this form of “deep” Polyak-Łojasiewicz has not appeared in any prior work.

---

> > ### Author Response · Authors · 2025-01-27
> >
> > Dear Reviewer 6E7M,
> >
> > We would be delighted to engage further with you in discussions. Moreover, please see our summary of proposed changes in our global comment above.
> >
> > Authors

---

> ### Comment · Reviewer_6E7M · 2025-02-01
>
> 1. Are the claims made in the submission supported by accurate, convincing, and clear evidence?
>
> From a theoretical standpoint, I do not find the paper’s treatment of Assumptions 1–2 sufficiently justified. While these assumptions may be reasonable at initialization, the manuscript does not provide a rigorous mathematical argument to show they continue to hold (or hold approximately) during the iterative training process. The absence of such justification undermines the clarity and accuracy of the paper’s theoretical claims. I recommend that the authors supply a more convincing analysis and discussion to show why Assumptions 1–2 remain valid throughout training.
>
> 2. Would some individuals in TMLR’s audience be interested in the findings of this paper?
>
> Despite the theoretical concerns above, the paper introduces a new framework and algorithm for deriving optimization methods geared toward non-convex composite objective functions—a topic likely to attract interest among researchers in optimization and machine learning. Hence, I do believe that some in the TMLR audience would find these ideas intriguing and worthy of further exploration.

---

> > ### Author Response · Authors · 2025-02-01
> >
> > Dear Reviewer 6E7M, thank you for taking the time to engage further!
> >
> > The paper is not meant to claim that all the assumptions hold throughout the iterative training process. For instance, we refer to Assumption 1 as a "simplifying assumption" that is used "for convenience". **If you could point to any sentence in the paper stating that Assumption 1 holds throughout the iterations, we would like to update the manuscript to correct this.**
> >
> > We see the overall flow of the paper as:
> > - we build a smoothness model for deep learning that is derived from the actual loss function under certain clearly-stated simplifying assumptions
> > - we derive an optimiser from this smoothness model
> > - we show that the optimiser has interesting properties
> >
> > In fact, we see working to lift the assumptions as being a really good direction for future work. As an example, the recent [Muon optimizer](https://kellerjordan.github.io/posts/muon/) can be seen as removing Assumption 2 and just working directly in a spectral norm geometry. Assumption 1 can be removed if we build the network entirely out of smooth components, such as by using GELU or SWIGLU nonlinearities, which are becoming more popular than ReLU nonlinearities nowadays.
> >
> > Again, **if you believe that the paper is making unsupported technical claims, we ask that you point to specific statements in the paper so that we may address them.**

---

> > > ### Comment · Action_Editor_8bW7 · 2025-02-02
> > >
> > > Let me ask this question in this thread because this might be related to the discussion here:
> > >
> > > at what $\boldsymbol{w}$ should the equation in Assumption 1 be evaluated? Do we need the condition to hold for all $\boldsymbol{w}$ in the provided proofs?

---

> > > ### Comment · Reviewer_6E7M · 2025-02-03
> > >
> > > I can treat the equation in Assumption 1 as defining a set for the weights $\Delta w$. I refer to this equation as Equation (0.0).
> > >
> > > Now, Theorem 1 can be understood as when Equation (0.0) holds, the new expansion can be obtained. I denote the key equation in Theorem 1 as Equation (0.1). In this sense, Corollary 1, Corollary 2, Lemma 5, and Theorem 2 all depend on Equation (0.1).
> > >
> > > Let $W^t$ denote the weight matrix at iteration $t$. The proofs of Lemma 8 and Theorem 3 require that the entire training trajectory satisfies Assumptions 1 and 2; that is, for each $W^t$, both Assumption 1 and Assumption 2 must hold. In particular, Assumption 1 holds with $\Delta W_k$ defined in Theorem 2, evaluated at $W^t$. This is why I argue that Assumption 1 needs to hold throughout the iterations.
> > >
> > > I personally disagree with the statement, ``But it turns out to be a good assumption in practice,'' regarding Assumption 2. The explanation for why a gradient-based algorithm would maintain Assumption 2 approximately true throughout the entire training process is not clearly discussed.
> > >
> > > Only when the authors' Assumptions 1, 2, and 3 hold (or approximately hold) throughout the iterations will the proof of Theorem 3 and some other results be valid.

---

> > > > ### Author Response · Authors · 2025-02-03
> > > >
> > > > Dear Reviewer 6E7M, we are sincerely grateful for you engaging further with us on this.
> > > >
> > > > To clarify, we are distinguishing between:
> > > > - claiming that the assumptions **actually hold in practice** throughout the iterations---**we ARE NOT doing this**
> > > > - deriving theoretical statements that depend on the assumptions holding throughout the iterations---**we ARE doing this**
> > > >
> > > > Since the theorem statements clearly mention that the results only hold under the assumptions, we maintain that **you are not actually pointing out factual inaccuracies in the theoretical results**. We feel that the issues you are raising can be dealt with simple re-wordings and clarifications to the text.
> > > >
> > > > Regarding the sentence "But it turns out to be a good assumption in practice", we will remove this sentence from the paper. Instead, we will just point out that this assumption has been observed empirically in Figure 1 of [Yang et al, 2023](https://arxiv.org/abs/2310.17813).

---

> ### Author Response · Authors · 2025-02-02
>
> Dear AE---thanks again for looking into this carefully.
>
> The point we are trying to make is that *"assuming X"* does not mean *"we claim that X is really true in practice"*. To make an analogy, if we wrote in the paper *"let's assume a spherical cow model, and analyse the system based on the assumption that cows are spherical"* this does not mean we are claiming that *"real cows are actually spheres"*. We hoped that reviewers would look at our paper and ask *"how does this model compare to existing models of cows"* and *"does this paper suggest ways to build even more accurate models of cows"*. But instead, the reviewers are coming back and saying *"the authors need to prove that cows are actually spheres for us to accept this paper"*. We do not claim that real life cows are actually spheres!
>
> To answer your question: yes some of the proofs rely on Assumption 1 holding along the iterations. But no, this does not mean we are claiming it would actually hold in practice. Assumption 1 is part of our model. And again, we know how to relax Assumption 1 for models built out of smooth components, which we plan to clarify in the manuscript.

---

### Review · Reviewer_fEb5 · 2024-11-09

**Summary Of Contributions:**

The authors introduce automatic gradient descent (AGD), which initializes the network and sets the learning rate as a function of architectural-specific properties like width and number of layers in order to be independent of the standard hyper-parameters in deep learning optimization setup. While AGD is primarily introduced for a full batch setting, the empirical analysis suggests that its stochastic version achieves comparable performance to that of SGD and Adam.

**Audience:**

Yes

**Claims And Evidence:**

Yes

**Requested Changes:**

- I would suggest moving some of the empirical results from the appendix to the main paper since they seem quite important for supporting the claims made in the introduction. (maybe reduce the sizes of Fig 1, 2). I would also suggest expanding/discussing the results more in section 4.
- I think standard deviation might be missing in some of the plots.
- I would also like the authors to address the weaknesses/questions raised in the *Strengths And Weaknesses* section.

**Strengths And Weaknesses:**

Strengths:
- The paper presents an interesting training framework, AGD, which incorporates architecture properties to initialize the model and regulate the step size. This can be quite crucial, as extensive hyper-parameter searches can often require a significant amount of computational resources.
- The theoretical analysis and background provided in the paper are thorough and well-written. For example, the authors connect their analysis to existing optimization frameworks, showing that existing methods can be recovered using Theorem 1.
- The authors did a great job in deriving and intuitively explaining their methodology. The convergence analysis for the proposed framework is also provided.

Weaknesses:
- The authors mention that AGD performance mainly falls below the well-tuned baseline. However, with default hyper-parameters (Fig. 3), AGD performs well on the FCN model but performs worse than Adam (for lr=1e-3) on a more popular ResNet-18 model. I wonder if the authors could provide more evidence where AGD outperforms default Adam/SGD on similar setups.
- While several existing works aim to reduce the sensitivity to hyper-parameters (some of them are cited by the authors in Table 1), the experimental baselines in this paper are limited to SGD and Adam.

Questions:
- The authors state that the inclusion of Assumption 1 did not lead to a weakening of the resulting algorithm in practice. I wonder if the authors could expand on it.
- What is the overall computational time required to fully train a model with AGD? Is it significantly less than conducting a small grid search over learning rates for SGD/Adam to find the optimal setup?
- Is there a connection between the automatic learning rate and the existing learning rate schedulers in optimization literature? The automatic learning rate as seen in Figures 4 and 6 appears to increase and then decrease, which could be similar to having a warmup followed by a decay.
- The authors state that the update is scaled relative to the norm of the weight matrix (Theorem 2). Could they provide clarification on how this is achieved, as it seems that the automatic step size primarily depends on the norm of the gradient matrix apart from architectural properties?

---

> ### Author Response · Authors · 2025-01-02
>
> Dear Reviewer fEb5, thanks a lot for your review! We are glad that you liked the theoretical analysis and methodology in our paper.
>
> Regarding other works on hyperparameter sensitivity, we have looked at some of these works and could immediately see flaws in them in contrast to AGD. For example, in Appendix D we showed that for the Prodigy optimizer, which claims to be parameter free, the training dynamics become unstable at large width. We believe that the root cause for this is that the Prodigy optimizer is not grounded in an optimization theory that accounts for neural architecture, like the one that we are trying to build. We also show on page 9 how our approach theoretically reconciles several different techniques in the literature. But with all that said, we see the role of this paper as trying to build a framework for deep learning optimization from the ground up. While certainly valuable, we leave extensive optimizer comparisons to future work.
>
> **Thank you for your interesting questions:**
> 1. We tried a variant of AGD where the derivation used an upper bound of the model linearisation error valid for deep linear networks instead of Assumption 1. We found that the variant did not perform better than the version presented in the paper. This might indicate that a learning rate based on a full majorization is too conservative.
> 2. The answer to your second question depends on the quality of your prior knowledge of a suitable Adam or SGD learning rate, and on your target accuracy. Depending on these factors, there will be situations where AGD is better and other situations where Adam / SGD is better.
> 3. We are not confident of a connection between the automatic learning rate and existing learning rate schedulers, although a good research goal would be to nail down such a connection.
> 4. When we say that the update is scaled relative to the norm of the weight matrix, we mean that the spectral norm of both scales with width like sqrt(fan-out / fan-in) under assumption 2. If one does not wish to make assumption 2, then one can modify the update rule to divide by the spectral norm of the gradient instead of the Frobenius norm, and then the statement will still hold.

---

> > ### Author Response · Authors · 2025-01-27
> >
> > Dear Reviewer fEb5,
> >
> > We would be delighted to engage further with you in discussions. Moreover, please see our summary of proposed changes in our global comment above.
> >
> > Authors

---

### Review · Reviewer_LKDi · 2024-12-23

**Summary Of Contributions:**

This paper tries to resolve a key problem in deep-learning that all well-functioning optimization algorithms have a lot of hyperparameters whose good values cannot be known in any principled way. To remedy the situation the authors propose a new heuristic called "Automatic Gradient Descent" (Algorithm 1 on page 8) which has no such parameters but is instead a adaptive gradient algorithm whose step-length is entirely decided by the current weights and the architecture of the net. The authors provide promising evidence for the algorithm but the theoretical justification is entirely unconvincing - and what makes the paper essentially untenable.

Its to be noted separately that the authors introduce this idea of functional expansion in Theorem 1 which seems critical to motivating AGD but the idea of functional expansion seems to have no clear meaning. If this thing has to be in the final paper then there is an urgent need of analysis of the consequential properties of a neural net or any predictor when it satisfies the equality given in Theorem 1. In this context it is to also be noted that the authors have frequently used the term "functional majorization" but this has never been formally defined. Section 2 of the paper needs to be entirely rethought if the authors opt for Option 1 of the requested changes suggested below.

**Audience:**

Yes

**Claims And Evidence:**

No

**Requested Changes:**

> Requested Change (OPTION 1)

If there has to be this convergence analysis in the accepted version of the paper then I strongly feel that either of the two things must happen,

- EITHER the authors must provide an example of a neural net class and a corresponding loss where all the 3+1 assumptions are provably true **and** either be able to prove Lemma 8 without assuming satisfaction of the three criteria at every iterate or show the proof of these 3 criteria to hold at every iterate of AGD.

- OR the paper needs to include an extensive experimental study showing that these assumptions (and additional conditions needed in Lemma 8 and Theorem 3) hold true in various neural net loss functions and while training them via AGD. One can start with demonstrating this on architectures already considered and also necessarily include some examples from each of the modern setups like PINNs, DeepONets, FNO, BERT, ViTs, GPTs, diffusion models, VAEs etc. A full scale of evidence for the assumptions is required if one cant produce a provable example of satisfaction of the assumptions and the proof conditions.

> Requested Change (OPTION 2)

Or I would suggest the authors to entirely drop this attempt at any a priori justification of AGD and write the paper solely focused on experimental evidence in favour of AGD - showing a full scale of evidence necessarily including many kinds of modern setups like, PINNs, DeepONets, FNO, BERT, ViTs, GPTs, diffusion models, VAEs etc. For almost all these setups instances are available which would run on even the free version of Colab and hence GPU requirements should likely not be a hurdle to execute this plan.

**Strengths And Weaknesses:**

This could have very well been a paper doing exhaustive experimental study of this AGD proposal by testing it in all kinds of benchmarks.

But the seriously floundering attempt to theoretically ground the algorithm makes it impossible to accept the paper as is.

This work needs a complete overhaul.

The theoretical attempt in this paper rests on a bunch of assumptions titled ``Prescription 1", "Assumptions 1 and 2".

Eventually the convergence proof is essentially a triviality because it assumes all the 3 assumptions to be holding at every iterate of the algorithm.

On top of these, there is a further "Assumption 3" that is needed to make Theorem 3 work!

---

> ### Author Response · Authors · 2025-01-02
> **Clearly stated assumptions can be a strength and not a weakness**
>
> Dear Reviewer LKDi, thank you for your review.
>
> We strongly disagree with the idea that a paper with clearly stated assumptions is untenable unless it also contains exhaustive empirical validation or all the assumptions are proved. We believe that our paper has substantial value in laying out a clear analytical skeleton for neural network optimizer design that shows how to account for nonlinear information about the model family through Lemma 4. While there are gaps in the skeleton, these gaps are clearly marked. This makes it easier for ourselves and other researchers to refine and extend the skeleton in future work. The other reviewers appreciated this aspect of the work, writing that “the authors did a great job in deriving and intuitively explaining their methodology” and that “the step-by-step development…is well-presented and accessible”.
>
> In short: the intention of our paper is not to do advanced math. The purpose of our paper is to build a conceptual framework for deriving optimizers for neural nets. Until we have a complete understanding of the problem, all work of this sort will necessarily contain gaps. The fact that we clearly state our assumptions is a strength and not a weakness, as it makes it easier for other researchers to build on and improve the work.
>
> ### **Response to requested changes:**
>
> **Option 1:** we are not suggesting that our assumptions are incontrovertible truths. The assumptions are supposed to highlight important quantities and explore how things work if we assume something about them. We propose to update the abstract to clearly indicate this. We provide further discussion of the assumptions in our response to Reviewer 6E7M, and we propose to also include this discussion in the manuscript.
>
> **Option 2:** this request seems to suggest abandoning the whole point of the paper, which is to try to build an analytical framework for designing deep learning optimization algorithms.

---

> > ### Comment · Reviewer_LKDi · 2025-01-25
> > **Thanks**
> >
> > I read the comments.
> > Without any of my requested change paths implemented I am not able to change my evaluation.

---

> > > ### Author Response · Authors · 2025-02-01
> > >
> > > Dear Reviewer LKDi,
> > >
> > > According to Oxford Languages, the definition of the word assumption is **"a thing that is accepted as true or as certain to happen, without proof."** While an analysis that makes assumptions may not be to the reviewer's taste, we feel that your review does not actually point out factual inaccuracies about the claims in our paper.
> > >
> > > Also, we feel that our work is valuable in starting to build a conceptual framework for integrating architectural information into deep learning optimization algorithms. **While there are gaps in the current framework, we flag these gaps clearly**. And we view working to lift the assumptions as a fantastic direction for future work.

---

### Author Response · Authors · 2025-01-02
**Global response**

We are grateful to all reviewers for their time and effort. We are keen to engage in discussions with the reviewers. We have requested an extension to the discussion period from the action editor because of the holidays. In this global response, we wish to address what we see as a major difference between ourselves and two of the reviewers in terms of how we think about the role that assumptions play in frameworks for optimizer design.

As we mention on page 6 of the paper, prominent optimization frameworks such as the Gauss-Newton method and natural gradient descent make strong assumptions. They assume that the model can be linearized when deriving the optimization step, thus neglecting non-linear effects in the model. Of course, we can still apply the Gauss-Newton method or natural gradient descent to nonlinear model families. Even when the linearity assumption does not literally hold, the Gauss-Newton method and natural gradient descent still provide us with a formal procedure for optimizer design. In fact, the nonlinear model setting is precisely the realm that these frameworks are intended for!

We see our framework as an attempt to improve upon natural gradient descent and the Gauss-Newton method. Our framework provides a formal procedure for designing optimization algorithms where information about the nonlinear structure of the model architecture is explicitly injected into the optimizer. For neural networks, this includes an understanding of how updates to different layers interact (Lemma 4) which we see as a major technical innovation. While our framework does make certain assumptions, the resulting optimizer can clearly still be applied in situations when these assumptions do not hold. Given the above context, we view the way we clearly state our assumptions as helping future research, which is a strength of our paper. We provide more specific discussion of the assumptions in our response to Reviewer 6E7M.

Overall, we believe that we have made significant progress by showing how to inject nonlinear architectural information into optimization algorithms. This should become increasingly important with the move to train ever deeper and more nonlinear neural networks that is happening in industry. While our framework is not the end of the story, it makes significant progress and suggests a number of exciting directions for future research.

---

### Author Response · Authors · 2025-01-27
**Summary of proposed changes**

Dear reviewers,

We want to re-affirm our belief that there is clear value in setting up a systematic framework for reasoning about deep learning optimization in an architecture-dependent way. Clearly stating our assumptions makes it easier for future work to build on and improve the framework.

To summarise our proposed changes:

1. move empirical results from the appendix to the main paper (Reviewer fEb5)
2. move convergence analysis (Section 3.2) to appendix (Reviewer LKDi)
3. clarify points raised by Reviewer fEb5
4. clarify assumptions (all reviewers)
- **Assumption 1:** the origin of this assumption is that we really need model linearisation error to behave to leading order like $O(\lVert \Delta w\rVert^2)$ to get a workable majorization. But for ReLU networks, which are non-smooth, the model linearization error can at worst behave to leading order like $O(\lVert \Delta w\rVert)$. To get around this issue, we made the simplifying assumption that we could ignore this term. But, as mentioned in the paper, there are other possible approaches. One would be to analyse deep linear networks, where we can prove that the model linearization error behaves like:
$$ \lVert f(w + \Delta w; x) - [f(w;x) + \nabla_w f(w;x)^\top \Delta w]  \rVert_2 \leq \left[\prod_{l=1}^L \left(1 + \frac{\lVert\Delta W_l\rVert}{\lVert W_l\rVert}\right) - 1 - \sum_{l=1}^L \frac{\lVert\Delta W_l\rVert}{\lVert W_l\rVert}\right] \times \left[\prod_{l=1}^L \lVert W_l \rVert\right] \times \lVert x \rVert_2,$$
where the matrix norms are the spectral norm. Another approach would be to analyse a smooth nonlinearity such as GELU, instead of ReLU. Overall, this should exemplify that while we made one decision, there are various other approaches that could be taken, and this could be a rich source of inspiration for future work.

- **Prescription 1:** This deals with the fact that, without some control on various norms of quantities inside the network (such as weights and inputs), a network will not exhibit a global majorization. For example, as the norm of a weight matrix grows, the network becomes increasingly sensitive to perturbations to the other layers. In this paper, we deal with this issue by assuming various quantities are of known and fixed size. This could inspire future work to consider projected gradient descent algorithms or regularization strategies designed to maintain these properties.

- **Assumption 2:** This assumption deals with basic conditioning properties of the gradient. While [other work has presented evidence for this assumption](https://arxiv.org/abs/2310.17813), the assumption could easily be removed and we would instead see the spectral norm instead of the Frobenius norm appearing in the analysis. In fact, this observation is currently inspiring interest in optimization algorithms for deep networks grounded in spectral norm geometry.

- **Assumption 3:** This assumption is mentioned simply to illustrate a setting where convergence to critical points implies convergence to global minima. We are not suggesting that this assumption actually holds in practice. The point is to write down a Polyak-Łojasiewicz-style inequality that reflects the actual layered structure of the neural network. This invites the reader to think about how neighbouring layers need to interact for such an inequality to hold. To the best of our knowledge, this form of “deep” Polyak-Łojasiewicz has not appeared in any prior work.

Overall, we feel that deep learning is in need of basic mathematical modeling work. While this paper is not the end of the story, it does make important progress.

---

### Comment · Action_Editor_8bW7 · 2025-02-02
**Assumption 1 compared to those in previous work**

Dear Authors,

Regarding Assumption 1, the manuscript says "this assumption is considerably milder than the common assumption in the literature (Pascanu & Bengio, 2014; Lee et al., 2019) that the model linearisation error is itself zero." Could you point me to the relevant parts of the references to verify this claim?

---

> ### Author Response · Authors · 2025-02-02
>
> Dear AE,
>
> Thanks for looking into this.
>
> Let's start with [(Pascanu & Bengio, 2014)](https://arxiv.org/abs/1301.3584). The relevant section of the paper is the analysis between Equation 3 and Equation 5, starting on page 2. The authors build a smoothness model (Equation 5) that only involves the gradient of the loss $\nabla \mathcal{L} (\theta)$ and first derivatives of the model (appearing through the Fisher information matrix $F$ defined in Equation 1). In other words, the authors assume that all terms involving higher model derivatives vanish. To justify this, they write things like *"Assuming $\Delta \theta \to 0$, we can approximate the KL divergence by its second order Taylor series*" or simply *"$\mathcal{L}(\theta + \Delta \theta)$ [is approximated] by its first order Taylor series $\mathcal{L}(\theta) + \nabla \mathcal{L}(\theta) \Delta \theta$"*. In short, **there is a linearisation assumption is at the core of natural gradient descent**.
>
> Regarding [(Lee et al, 2019)](https://arxiv.org/abs/1902.06720), this is actually a main claim of their paper. The title of the paper is *"Wide Neural Networks of Any Depth Evolve as Linear Models Under Gradient Descent"* and in the abstract the authors write that the learning dynamics of infinitely wide neural networks *"are governed by a linear model obtained from the first-order Taylor expansion of the network around its initial parameters"*. In short, **there is a body of work claiming neural networks can be understood through linear models**.
>
> Even on the Wikipedia page for the [Gauss-Newton algorithm](https://en.wikipedia.org/wiki/Gauss%E2%80%93Newton_algorithm) for non-linear least squares, it's mentioned that convergence of the algorithm is not guaranteed unless, for instance, *"the functions are only "mildly" nonlinear"* such that the second derivatives of the model are dominated in magnitude by the first derivatives. In short, **there is a linearisation assumption at the core of the Gauss-Newton method.**
>
> We are not meaning to endorse these approaches here. We are just pointing out that it is completely standard in the literature to build smoothness models that assume the model being optimized is linear. Our paper shows how to relax these assumptions by including nonlinear model information through Lemma 4. We also clearly signposted our assumptions to make it easy to see how the analysis fits together and so future work can work on refining the approach---we actually regard this as a strength of our paper.

---

### Author Response · Authors · 2025-02-17
**Gentle nudge**

Dear Action Editor,

We just wanted to send a gentle nudge that it would be great to get a decision made on this.

Just to re-iterate, but our perspective on this paper is that we made substantial progress toward building a workable smoothness model for deep learning. This has been a major open problem in the field. While our paper relies on certain clearly stated structural assumptions, we believe that relaxing these assumptions provides a fantastic opportunity for future research and progress. And, in any case, making clearly stated assumptions in an analysis does not conflict with the TMLR acceptance criteria.

In fact, a lot of work downstream of our paper is starting to build on the ideas that we initiated here. For example:

- [this recent paper](https://arxiv.org/abs/2502.07529) builds Frank-Wolfe methods that maintain norm constraints through training (see insight 3.1). Therefore this paper provides a route to ensuring that Prescription 1 holds throughout training
- [this recent blog post](https://kellerjordan.github.io/posts/muon/) proposes an optimization method working directly in a spectral norm geometry, which can be seen as removing the need for Assumption 2 (we used Assumption 2 simply to switch from spectral norms to Frobenius norms)
- [this Jupyter notebook](https://docs.modula.systems/examples/weight-erasure/) validates the claim that the gradients can be low stable rank independent of batch size (see the figure at the bottom), but also shows that algorithms working directly in a spectral norm geometry converge faster

If the paper is accepted, we plan to update the paper and revise the presentation to address the reviewer concerns.

---

### Decision · Action_Editor_8bW7 · 2025-02-15

**Recommendation:** Reject

**Comment:**

A reviewer's recommendation is "leaning accept" and the other two recommends rejection. The positive comments from the reviewers include the proposed framework is a novel and can have a great impact on the efficiency of practice of training deep neural networks. The main reason for the recommendations of rejection is the lack of evidence on how relevant the assumptions are. Adding empirical studies or theoretical analyses on how the proposed update rule can maintain such properties. I hope the authors do not take this recommendation as a discouragement because the reviewers also expressed positive opinions on the overall direction of this work, and it has a great potential that this work turns out to be an important breakthrough. The authors are encouraged to incorporate the feedback from the reviewers in the future revision and resubmit it to TMLR or other ML conferences.

**Audience:**

The reviewers (and myself) find the problem that this work addresses important and the approach interesting. The authors and two reviewers also mention that this novel framework may well inspire future research. It is likely that it interests many researchers of the general ML community.

**Claims And Evidence:**

This work proposes a novel first-order optimization method, called _Automatic Gradient Descent (AGD)_, that does not have hyper-parameters such as learning rate to be tuned.
To derive the update rule of the proposed method (Theorem 2 and Algorithm 1), the authors derive a majorization bound (Lemma 5 ) on the objective function, based on some assumptions (Assumptions 1-2, Prescription 1) related to the smoothness of the model in interaction with the loss function. The paper then presents analyses on convergence to a critical point using the same assumptions as well as convergence to a global minimum when an additional assumption of Polyak-Łojasiewicz inequality type (Assumption 3).

Moreover, the authors provide experiments using several deep neural networks and datasets to see how the proposed method performs and compares to Adam and SGD. In some cases, the proposed method shows performance superior to Adam and SGD with their default hyper-parameters and comparable to those with tuned hyper-parameters.

Under the assumptions, the theoretical results seem correct as the reviewers (and myself) did not find errors. The empirical claims are supported by the experiments.

However, whether the assumptions (and hence the theoretical results) are relevant depends on how the training trajectory given by the proposed method can maintain properties close to the assumptions (if not exactly). The current manuscript does not provide evidence supporting this point, as Reviewers LKDi and 6E7M express.

**Resubmission Of Major Revision:**

The authors may consider submitting a major revision at a later time.

---

> ### Author Response · Authors · 2025-02-17
>
> Thanks to the action editor and the reviewers for their time and effort in evaluating the work and providing helpful feedback!

---

> > ### Comment · Action_Editor_8bW7 · 2025-02-18
> >
> > I thank the authors and the reviewers for the valuable discussions. I wish the authors the best for their future submission on this work.